# Hypoxia-Induced Cancer Cell Responses Driving Radioresistance of Hypoxic Tumors: Approaches to Targeting and Radiosensitizing

**DOI:** 10.3390/cancers13051102

**Published:** 2021-03-04

**Authors:** Alexander E. Kabakov, Anna O. Yakimova

**Affiliations:** Department of Radiation Biochemistry, A. Tsyb Medical Radiological Research Center—Branch of the National Medical Research Radiological Center of the Ministry of Health of the Russian Federation, Koroleva 4, 249036 Obninsk, Russia; anna.prosovskaya@gmail.com

**Keywords:** radiotherapy, radioembolization, hypoxia-inducible factor-1, heat shock transcription factor 1, metabolic reprogramming, autophagy, cancer stem cells, epithelial–mesenchymal transition, epigenetic regulation, exosomes

## Abstract

**Simple Summary:**

Some regions of aggressive malignancies experience hypoxia due to inadequate blood supply. Cancer cells adapting to hypoxic conditions somehow become more resistant to radiation exposure and this decreases the efficacy of radiotherapy toward hypoxic tumors. The present review article helps clarify two intriguing points: why hypoxia-adapted cancer cells turn out radioresistant and how they can be rendered more radiosensitive. The critical molecular targets associated with intratumoral hypoxia and various approaches are here discussed which may be used for sensitizing hypoxic tumors to radiotherapy.

**Abstract:**

Within aggressive malignancies, there usually are the “hypoxic zones”—poorly vascularized regions where tumor cells undergo oxygen deficiency through inadequate blood supply. Besides, hypoxia may arise in tumors as a result of antiangiogenic therapy or transarterial embolization. Adapting to hypoxia, tumor cells acquire a hypoxia-resistant phenotype with the characteristic alterations in signaling, gene expression and metabolism. Both the lack of oxygen by itself and the hypoxia-responsive phenotypic modulations render tumor cells more radioresistant, so that hypoxic tumors are a serious challenge for radiotherapy. An understanding of causes of the radioresistance of hypoxic tumors would help to develop novel ways for overcoming this challenge. Molecular targets for and various approaches to radiosensitizing hypoxic tumors are considered in the present review. It is here analyzed how the hypoxia-induced cellular responses involving hypoxia-inducible factor-1, heat shock transcription factor 1, heat shock proteins, glucose-regulated proteins, epigenetic regulators, autophagy, energy metabolism reprogramming, epithelial–mesenchymal transition and exosome generation contribute to the radioresistance of hypoxic tumors or may be inhibited for attenuating this radioresistance. The pretreatments with a multitarget inhibition of the cancer cell adaptation to hypoxia seem to be a promising approach to sensitizing hypoxic carcinomas, gliomas, lymphomas, sarcomas to radiotherapy and, also, liver tumors to radioembolization.

## 1. Introduction

Radiation exposure is the significant modality to fight cancer, and most of patients with solid malignancies receive courses of photon or hadron therapy (reviewed in [1,2]). Unfortunately, antitumor radiotherapy is not always effective; in particular, the enhanced radioresistance is often exhibited by cancer cells within the hypoxic zones of tumors, and this is a great challenge [3,4].

Although tumor cell-stimulated angiogenesis is one of the hallmarks of cancer [5], the aggressive malignancies usually contain poorly vascularized regions where the angiogenic potential lags behind the superfast proliferation of malignant cells. Through the insufficient development and functional abnormality of the vasculature in some tumor loci, cancer cells residing within these loci undergo oxygen deficiency, i.e., hypoxia [3,4,5]. In terms of the relatively low density of capillaries and limited blood perfusion, the oxygen concentration in tumor tissue decreases as it moves away from the capillary with oxygenated blood (Figure 1). While acute hypoxia can repress the proliferative activity of cancer cells and even kill some of them, many hypoxia-stressed malignant cells are adapted by triggering a certain gene expression and signaling pathways aimed at the acquisition of a hypoxia-resistant phenotype [3,4]. The so-called cycling hypoxia is not restricted to the tumor regions with low vascular density, but this phenomenon can also affect the state and responses of the involved tumor cells [6,7]. Importantly, several hypoxia-responsive adapting mechanisms enhance the radioresistance of cancer cells undergoing hypoxia (see [3,4,6,7]; Figure 1), and this may negatively influence the outcome of radiotherapy. Furthermore, the same adapting mechanisms may render hypoxic cancer cells less susceptible to some chemotherapeutics and radiosensitizers derived from hypoxia-activated prodrugs. Accordingly, some inhibitors of those hypoxia-induced cellular mechanisms may be exploited for better targeting hypoxic tumors.

Moreover, the hypoxic regions of aggressive carcinomas are sites where the “epithelial–mesenchymal transition” (EMT) results in the emergence of invasive, mobile cells with a cancer stem cell (CSC)-like phenotype that are able to migrate and penetrate into the vasculature to form distant metastases, while being more chemo- and radioresistant than non-stem cancer cells (see [8,9], Figure 2 and Section 7). EMT-resembling phenotypic modulation may occur within the hypoxic niches of sarcomas, gliomas, lymphomas and other types of solid malignancies when the involved tumor cells increase their stemness, enhancing their motility, invasiveness, metastatic potential and chemo-/radioresistance. The exosome secretion by hypoxia-stressed cancer cells also promotes their migration, invasion and metastasis spread (see Figure 2 and Section 9). Such hypoxia-driven generation of the actively migrating radioresistant CSC-like cells is conducive to the weak responsivity of hypoxic tumors to radiotherapy, since blood circulation allows these vasculature-penetrating tumorigenic cells to flee the zone being exposed to therapeutic irradiation (Figure 2).

Besides the hypoxia-triggered cellular radioprotective responses, the lack of oxygen in the microenvironment of cancer cells per se enhances their radioresistance. The so-called “oxygen effect” is a phenomenon that is well-known among radiobiologists and implies the significantly decreased post-radiation survival of cells being exposed to ionizing radiation in the conditions of normoxia (20–21% O_2_) or hyperoxia (>21% O_2_) as compared with the consequences of radiation exposure performed under anoxia or hypoxia [10]. The oxygen effect-associated cell radiosensitivity is positively correlated with the oxygen concentration in the cell microenvironment at the moment of irradiation; this is usually explained by the oxygen-mediated “fixation of damage” when molecules of O_2_ are involved in reactions of radiation-induced free radicals (mainly the hydroxyl radicals) with cellular DNA, thus aggravating radiation damages to DNA and making these damages irreparable (and often lethal) for the target cell (see [10]; Figure 3). Consequently, the acute deficiency of oxygen in the poorly vascularized regions of tumors can, by itself, contribute to the radioresistance of cancer cells residing in such regions and be one of the causes of a low efficacy of radiotherapy [4,10,11]. If so, artificial oxygenation of the hypoxic regions in tumors would help to radiosensitize malignant cells that are otherwise radioresistant through their oxygen-deficient microenvironment. Alternatively, the use of high linear energy transfer (high-LET) radiation inducing, largely, oxygen-independent cancer cell elimination would allow radiotherapists to better target hypoxic tumors. Indeed, compared with the effects of low-LET (photon) radiation, the intensity of cell death resulting from the exposure of some fractions of protons (within the Bragg peak), carbon ions or α-particles (kinds of high-LET radiation) is much less dependent on the oxygen concentration in the cell microenvironment [2,4,11]. However, the hypoxia-adapted cancer cells have a certain phenotype with specific regulation that enables them to better resist both low- and high-LET radiation (Figure 1). That is why additional (radiosensitizing) cotreatments may enhance the efficacy of high-LET radiation-based therapy toward hypoxic tumors.

It should be noted that the hypoxic conditions in patients’ tumors can arise as a result of some therapeutic procedures. For example, antiangiogenic therapy is sometimes used against solid malignancies, and this may decrease the supply of oxygenated blood to tumor cells, thereby causing hypoxia in some tumor regions [5]. Antiangiogenic therapy can be combined with radiotherapy [5,12]; in those cases, the treatment-provoked hypoxia will enhance the tumor radioresistance so that a radiosensitizing cotreatment seems to be needed herein. Similarly, intratumoral hypoxia takes place under transarterial embolization with microspheres [13] that are used against liver-localized malignancies. In the case of radioembolization, when microspheres containing radioactive isotopes of Yttrium-90 or Holmium-166 embolize arterioles and capillaries supplying oxygenated blood to the target tumor [14], the treatment-created conditions of local hypoxia may render the involved cancer cells more radioresistant and, hence, impair the therapeutic benefit of the radionuclide irradiation. If so, the therapeutic efficacy of radioembolization may be improved by some additional modalities enabling to minimize the hypoxia-associated radioresistance of cancer cells.

Thus, the high radioresistance of hypoxic tumors is a serious challenge for anticancer radiotherapy, and resolving this challenge seems to be the actual task. Summarizing a large amount of data, the present review considers the molecular and cellular basis of the radioresistance of hypoxic tumors and also critically estimates some approaches to radiosensitizing cancer cells in hypoxic tumor regions. The authors of this review advance an idea about the necessity of combinative and multitarget treatments in order to overcome the radioresistance of hypoxic tumors.

## 2. Hypoxia-Inducible Factors: Their Regulation and Contribution to the Tumor Radioresistance

Hypoxia-inducible factor-1 (HIF-1) is a transcription factor that is the master regulator of cellular responses to hypoxia. Its molecule is a heterodimer formed by one of the three hypoxia-inducible subunits, such as HIF-1α or HIF-2α or HIF-3α, and an oxygen-insensitive (constitutively expressed) subunit HIF-1β. All HIF-1 target genes contain the specific nucleotide sequence of “hypoxia-responsive element” (HRE: 5′-(A/G)CGTG-3′), to which the HIF-1 heterodimer is bound to initiate their transcription; most products of HIF-1 target genes eventually serve for the local attenuation of hypoxic stress and preservation/adaptation of the involved cells (see [15,16,17]; Figure 4). Importantly, both HIF-1α and HIF-2α were shown to contribute to the high resistance of hypoxic tumors to chemotherapy and radiotherapy [15,16,17,18].

Due to its special domain organization and a set of cofactors regulating its (in)activation, subcellular localization and degradation, HIF-1α acts as an oxygen sensor whose expression, functional activity and stability are enhanced in response to hypoxia, whereas they are decreased under normoxia. Site-specific phosphorylation, deacetylation and, also, S-nitrosation at cysteine-800 of HIF-1α, the chaperoning of HIF-1α by heat shock proteins (HSP90 and HSP70) and the interaction of HIF-1α with the cAMP response element-binding protein (CREB)-binding protein (CBP)/p300 coactivator result in the nuclear translocation of this subunit, its dimerization with HIF-1β and the binding of this heterodimer to the HRE, followed by the transcription of HIF-1 target genes ([15,16,17,19]; Figure 4). In the conditions of normoxia, α-ketoglutarate- and O_2_-dependent prolyl-4-hydroxylases (PHDs) catalyze the hydroxylation of two proline residues (P402 and P564) within the so-called “oxygen-dependent degradation domain” (ODDD) of HIF-1α; such a modification ensures the subsequent polyubiquitination and proteasomal degradation of HIF-1α through the von Hippel-Lindau (VHL)-dependent proteolysis [15,16,17]. In parallel, asparagine-803 in HIF-1α is also hydroxylated, in an oxygen-dependent manner, by asparaginyl hydroxylase (factor inhibiting HIF-1: FIH-1), which prevents the HIF-1α–CBP/p300 interactions ([15,16,17]; Figure 4). Thus, there is the oxygen-sensitive mechanism regulating the stability, subcellular localization and functional activity of HIF-1α. Besides, certain protein kinases and phosphatases, histone lysine methylases and demethylases, acetyl transferases and deacetylases, chaperones and a small ubiquitin-related modifier (SUMO) E3 ligase can directly or indirectly affect the HIF-1α stability and its capacity to form the active HIF-1 heterodimer in the nucleus [19,20,21,22,23,24]. At the expression level, HIF-1α can also be regulated by a signal transducer and activator of transcription 3 (STAT3) [25], nuclear factor-κB (NF-κB) [26], microRNAs [15,27,28] or long noncoding RNAs [15,29,30,31,32], c-Myc [33], angiotensin II [34] and signal transduction pathways involving stress- or mitogen-activated kinases [35,36], phosphoinositide 3-kinases (PI3K) and the mammalian target of rapamycin (mTOR) [37,38,39].

HIF-2α has 48% amino acid homology with HIF-1α, and there is a similarity in the domain organization of their molecules [15,16,17]. Like HIF-1α, HIF-2α can form heterodimers with HIF-1β that bind to the HRE to trigger the transcription of HIF-1 target genes. In contrast to ubiquitously expressed HIF-1α, HIF-2α is mostly expressed in the vascular endothelium, lung and heart tissues and placenta; the transcriptional targets of HIF-1α and HIF-2α are somewhat different, although the expression of some hypoxia-inducible genes can be initiated by both HIF-1α and HIF-2α. The major products of HIF-1 target genes are angiogenic factors, glucose transporters and key glycolytic enzymes, carbonic anhydrase 9 and regulators of energy metabolism or apoptosis ([15,16,17]; Figure 4), so that the HIF-1-mediated cellular response is mainly aimed at the compensation for and adaptation to hypoxic stress. Notably, there is a nongenomic phosphorylation-dependent pathway involving HIF-1α in the protection of hypoxia-stressed cells from apoptosis [40].

### 2.1. HIF-1-Mediated Radioprotective Mechanisms in Hypoxic Cancer Cells

The involvement of HIF-1 in the radioprotective responses of hypoxic tumors seems to be multifaceted and can include the HIF-1-mediated influence on the cell cycle and energy metabolism, signaling pathways, EMT (i.e., generation of CSC-like cells), autophagy, DNA damage response, epigenetic factors, cytoprotection from apoptosis and oxidative stress, etc. [15,16,17,18]. In particular, the HIF-1α-conferred radioresistance of cervical carcinoma HeLa cells under simulated hypoxia was associated with the decrease in expression of Bax and p53 and protection against radiation-induced apoptosis [41,42]. The antiapoptotic activity of HIF-1α, being upregulated in response to simulated hypoxia, was revealed in human oral squamous cell carcinoma cells (Ca9–22 line) exposed to irradiation [43]. The hypoxia-induced radioresistance of prostate cancer cells (22Rv1 and DU145 cell lines) was connected with HIF-1α-mediated G2/M arrest and decreased apoptosis in the irradiated cells [44]. In hypoxia-stressed prostate cancer cells of two other cell lines (LNCaP and C4-2B), overexpressed HIF-1α was shown to ensure the enhanced β-catenin nuclear translocation that led to radioresistance due to an altered cell cycle, reduced apoptotic death and improved nonhomologous end joining in the post-radiation DNA break repair [45].

The hypoxia-associated expression of erythropoietin and its receptor in cancer cells is also regulated by HIF-1 [46], while erythropoietin signaling can contribute to the radioresistance of gliomas [47,48]. The erythropoietin receptor silencing in glioblastoma cells did stimulate their senescence and mitotic catastrophes after X-ray irradiation [48]. In another report [49], the hypoxia-induced radioresistance of glioblastoma cells was due to the HIF-1α-mediated expression of livin, an antiapoptotic protein, and this radioprotection can be blocked by the targeted inhibition of livin with the cell-permeable peptide TAT-Lp15 (TAT domain from human immunodeficiency virus-1 fused with the truncated peptide derived from linear peptides that specifically bind to livin).

The HIF-1α-dependent radioprotection of hypoxic gastric cancer cells appears to be realized via the HIF-1α-mediated expression of human epididymis protein 4 (HE4), whose upregulation is responsible for the high resistance of gastric cancer to radiotherapy [50].

Zhong et al. [51] described the HIF-1α-dependent expression of beclin with the subsequent activation of autophagy as events promoting the enhanced radioresistance of breast cancer cells (MCF-7 line) being exposed to hypoxia. Similarly, a contribution of HIF-1α to the hypoxia-induced expression of LC3-II and autophagy was suggested as the radioprotective mechanism acting in the hypoxic cells of lung cancer [52] and osteosarcomas [53]. The other interrelations between HIF-1α, autophagy and radioprotection were revealed in colon cancer cells in which hypoxia-induced autophagy reduced their radiosensitivity via the HIF-1α/miR-210/Bcl-2 pathway [54] and, also, in hypoxic hepatoma and U251MG glioma cells, whose radiosensitivity was enhanced after reducing autophagy through the HIF-1/Akt/mTOR/P70S6K pathway [29].

As for the hypoxia-associated reprogramming of the energy metabolism in cancer cells, HIF-1 is a major switcher in their ATP-generating pathways from the mitochondrial oxidative phosphorylation toward glycolysis (the Warburg effect) [55,56]. Such a switching is largely based on the HIF-1-dependent expression of key enzymes and regulators of carbohydrate metabolism and leads to the decrease in producing reactive oxygen species (ROS) and intracellular accumulation of reduced glutathione (GSH); all this potentiates both the antioxidant capacity of hypoxic tumor cells and their radioresistance (see Figure 4 and Figure 5 and Section 3). Interestingly, the HIF-1 signaling pathway is a targetable link between the cellular metabolism, circadian rhythm, DNA repair and response to radiotherapy in high-grade gliomas (reviewed in [57]).

At the level of epigenetic regulation, the HIF-1-induced expression of microRNA-210 (miR-210) was shown to promote the enhanced radioresistance in hypoxic cells of lung cancer cell lines A549 and H1975; this was manifested in the decreased apoptotic death, improved growth and accelerated repair of the DNA double-stranded breaks in the irradiated cancer cells exhibiting the “miR-210 phenotype” [58]. Under hypoxia in colon cancer cells, the HIF-1α-dependent expression of miR-210 ensured the radioprotective mechanism involving autophagy and Bcl-2 [54]. At the same time, miR-210 is known to stabilize HIF-1 [59], so that there is a positive regulatory loop defining the HIF-1-based radioresistance of hypoxic tumors. In an ovarian cancer model, HIF-1α suppressed miR-299 in the hypoxic cancer cells that were associated with the increased expression of heparanase HPSE1 and an elevated radioresistance [60]. In turn, long intergenic noncoding (linc) RNA-p21 promotes the intracellular accumulation of HIF-1α by reducing its ubiquitination; this linc RNA-mediated stabilization/accumulation of HIF-1α confers the radioresistance of hypoxic hepatoma and glioma cells through the induction of autophagy and the antiapoptotic Akt/mTOR/P70S6K pathway [29].

Importantly, HIF-1 is involved in the EMT induction in hypoxic niches and maintenance of the cancer stemness [61,62] that increases the pool of radioresistant CSC-like cells and, hence, impairs the response of hypoxic tumors to radiotherapy. It was revealed that HIF-1 in cancer cells surviving radiation exposure promotes their translocation toward blood vessels [63]. This HIF-1-dependent mechanism appears to allow a part of cancer cells to leave the site undergoing serial therapeutic irradiation to form distant metastases in other tissues and organs, therefore nullifying the outcome of radiotherapy (Figure 2). Wang et al. [64] reported that HIF-1α is required for the hypoxia-induced development of a stem phenotype in Hep-2 human laryngeal squamous carcinoma cells that is accompanied by the enhancement of radioresistance. In a model with laryngeal squamous carcinoma CNE-2 stem cells, the NF-κB/HIF-1 signaling pathway was shown to maintain a cancer stemness and high radioresistance in CD133-positive CSCs, whereas the inhibition of this HIF-1-involving pathway reversed the EMT and diminished the radioresistance [26]. It is HIF-1α that largely defines the radioresistance of CSCs of head and neck squamous cell carcinoma to both photon and carbon ion irradiation [65].

All this means that HIF-1 participates in several radioprotective mechanisms acting in hypoxic tumors. Besides the above ones, there are other HIF-1-based mechanisms contributing to hypoxia-associated tumor radioresistance that may be suppressed by various inhibitors (see Table 1).

### 2.2. Targeting HIF-1 to Sensitize Hypoxic Tumors to Radiation Exposure

Many studies with the knocking down of HIF-1α and HIF-2α in cancer cells have proven the significant role of these factors in promoting malignant growth and tumor resistance to hypoxia, as well as chemo- and radiotherapy (reviewed in [15,16,17,18]). Therefore, HIF-1α and HIF-2α seem to be therapeutically significant targets for the chemo- and radiosensitization of hypoxic tumors; all the more, the domain organization of HIFs and multiple levels of their expression/activity regulation imply a lot of possibilities for inhibitory targeting. The cancer-promoting activities of HIF-1 can be suppressed by various agents either directly interacting with HIF-1 molecules or targeting pathways that regulate HIF-1 via microRNAs, transcriptional factors, chaperones, protein kinase cascades, etc. (see [16,100] for a review). A number of small molecule inhibitors of the functional activity or expression of HIF-1 have been described that exerted hopeful radiosensitizing effects in different models related to tumor hypoxia (see Table 1).

The agents listed in Table 1 are only the ones that were shown to inhibit the HIF-1-associated radioresistance of tumor cells in the model experiments. Since HIF-1 is thought to be a significant target for chemotherapy as well, various compounds targeting HIF-1α and/or HIF-2α have been developed and tested as potential anticancer drugs in (pre)clinical trials [100]. In silico modeling may, in turn, assist in the creation of novel HIF-1 inhibitors and evaluate their potential effectiveness, toxicity and bioavailability [4].

The data from Table 1 provide the proof-of-principle that the artificial inhibition of HIF-1 can sensitize hypoxic tumors to radiotherapy. Importantly, such an approach may also work in the case of specific methods of radiation exposure such as immunoradiotherapy [71] and neutron capture therapy [78]. It seems likely that HIF-1 inhibition in the target tumor prior to its treatment with radioembolization would improve the therapeutic outcome. One can wonder why the problem of the radiosensitization of hypoxic tumors is not resolved so far, if among the revealed HIF-1-targeting radiosensitizers there are nelfinavir [66], atorvastatin [70,71], temsirolimus [39], paclitaxel [93], bortezomib [94,95], sorafenib [96] and sunitinib [97], which are already applied in clinical practice. Furthermore, the latter five are remedies of anticancer chemotherapy and may be combined with radiotherapy. However, all is not so easy herein, and the in vivo situation may be complicated by a number of endogenous factors. In particular, the severe and prolonged dysfunction of HIF-1 will aggravate hypoxia in target tumors through the full blockade of angiogenesis. If so, enhanced radiosensitization may be achieved under the successive combination of HIF-1 inhibition and artificial oxygenation of the hypoxic tumor: the former would impair the HIF-1-based adaptive and radioprotective mechanisms that were driven by chronic hypoxia before irradiation, while the latter would augment the killing of the irradiated cancer cells due to the “oxygen effect”. Such a combinative treatment may yield synergistic effects thanks to the oxygen-responsive degradation of HIF-1α via the VHL-dependent proteolysis (but not in the case of the application of bortezomib, which inhibits proteasomes). This is exactly what was achieved by Iijima et al. [101], who used oxygen nanobubbles to radiosensitize lung cancer EBC-1 cells and breast cancer MDA-MB-231 cells: the intracellular levels of HIF-1α were reduced through its degradation resulting from oxygenation, while the oxygen effect-associated radioresistance was overcome. Interestingly, such oxygen nanobubbles can be loaded with a cytotoxic drug (or a radiosensitizer) to simultaneously reverse hypoxia, suppress HIF-1α-mediated cytoprotection and induce cytotoxicity in target tumors, as was shown when using oxygen nanobubbles loaded with doxorubicin [102].

The reoxygenation of target (hypoxic) tumors may lead to other complications related to HIF-1. In particular, the post-radiation generation of ROS in reoxygenated tumors was shown to result in the nuclear accumulation of HIF-1 and the increased production of HIF-1-regulated cytokines, along with the radioprotection of vascular endothelial cells [103]. Inhibiting this ROS-responsive activation of HIF-1 in irradiated and reoxygenated tumors can radiosensitize them via the enhanced destruction of tumor vasculature [103,104]. An S-nitrosation-dependent mechanism of the stabilization/activation of HIF-1α under normoxia has been discovered in irradiated murine tumors [105]. In contrast to the S-nitrosation of HIF-1α at cysteine 800 that takes place under hypoxia and is required for HIF-1-mediated transcription (reviewed in [16]), the S-nitrosation of HIF-1α at cysteine 533 can occur in normoxic tumors undergoing radiation exposure [105]. Nitric oxide, being generated by tumor-associated macrophages in response to irradiation, modifies cysteine 533 in the ODDD of HIF-1α, which preserves its molecule from oxygen-dependent degradation and, thus, ensures the functional activity of HIF-1α under normoxia [105]. Such an HIF-1α-involving mechanism may decrease the radiosensitizing effects under the reoxygenation of hypoxic tumors. Therefore, combining radiation exposure and a HIF-1 blockade seems rational toward reoxygenated tumors.

The other problem is the delivery of a HIF-1-targeting radiosensitizer into cancer cells residing within the hypoxic loci. The effectiveness of drug transport from blood to solid malignancies depends on many factors and is restricted by the abnormal organization/characteristics of tumor vasculature and extravascular tumor tissue (see [106] for a review). Due to poor vascularization or transarterial embolization by microspheres, pharmacological agents injected into the blood flow are poorly delivered to the hypoxic regions of target tumors. Moreover, some hypoxia-adapted cancer cells (including CSC-like cells in hypoxic niches) overexpress membrane transporters such as ATP-binding cassette subfamily B member 1 (ABCB1) and ATP-binding cassette subfamily G member 2 (ABCG2), which pump xenobiotics out the cell and confer the so-called multidrug resistance [107]. Thanks to these transporters, at least some subpopulations of tumor cells in the hypoxic zone are able to actively exclude small-molecule radiosensitizers (e.g., paclitaxel) and, hence, remain radioresistant. It seems possible, though, that some inhibitors of HIF-1 are not pumped out by the membrane transporters, and this possibility should be examined under preclinical trials of either radiosensitizer. Some of the HIF-1 inhibitors may downregulate the pumping-drug-out function of the membrane transporters in hypoxia-adapted cancer cells and CSC-like cells; in this respect, ursolic acid seems attractive, because this compound is an inhibitor of the HIF-1α expression [82] and able to downregulate ABCG2 in CSCs [108]. Noteworthy is the effect of sorafenib, which, in combination with γ-irradiation, suppressed the HIF-1 expression and killed CSCs in breast cancer cells (MDA-MB-231 and MCF-7 lines) subjected to hypoxia [96]. Such a promising combination (sorafenib + irradiation) should be examined in in vivo models of hypoxic tumors; notably, sorafenib may upregulate HIF-2α in cancer cells [109].

Alternatively, the use of special microcarriers on the basis of liposomes or nanoparticles or dendrimers containing pharmacological inhibitors of HIF-1 or oligonucleotide vectors downregulating the HIF-1α expression may be an effectual way to suppress HIF-1 inside hypoxic tumor cells. In this respect, encouraging data were reported by Zhou et al. [91], who succeeded in the radiosensitization of hypoxic tumor cells in a murine model of breast cancer using yolk-shell Cu_2-x_Se@PtSe nanoparticles (a nanosensitizer) with acriflavine (an HIF-1α inhibitor) on their surface. Among the beneficial effects achieved in the target tumor, there were the reduced expression of HIF-1α and VEGF, lower microvessel density, endogenous generation of O_2_ from H_2_O_2_ (i.e., artificial oxygenation overcoming the oxygen effect), enhanced production of ROS, cell cycle arrest, additional DNA double-stranded breaks and increased apoptosis after X-ray irradiation—all this allowed the authors to talk about the synergistic enhancement of the radiation response in hypoxic cancer cells [91]. An analogous nano approach may be applied for the development of HIF-1 inhibition-based modality aimed at overcoming hypoxia-induced radioresistance in human tumors.

In some cases, an additional problem may be the presence of concomitant ischemic pathology in cancer patients, because HIF-1 inhibition may aggravate the ischemia-induced injury of an affected organ. Such a situation requires the in vivo selectivity in either action or delivery of HIF-1 inhibitors in order to effectively target hypoxic tumors without a risk of fatal infarction. Hypothetical HIF-1 inhibitors produced by hypoxia-activated or low pH-sensitive prodrugs will, probably, not be suitable, as they will act not only within the hypoxic loci of tumors but, also, in ischemic zones of the myocardium or brain. Nevertheless, intratumoral HIF-1 seems to be the extremely attractive molecular target for sensitizing hypoxic tumors to radiotherapy.

## 3. Hypoxia-Induced Reprogramming of Energy Metabolism

### 3.1. Energy Metabolism in Hypoxic Tumor Cells and How It Is Linked to Their Radioresistance

Cancer cells residing in the poorly vascularized or embolized tumor region have to adapt to the conditions of chronic oxygen deficiency and restricted nutrient availability; these cells modulate their energy metabolism toward the overwhelming predominance of glycolysis (the Warburg effect), along with the more active uptake of glucose [110,111]. This forced switch to an anaerobic way of ATP generation is accompanied by repressing both the mitochondrial respiratory function and many oxygen-consuming reactions in hypoxia-stressed cancer cells, which acquire some advantages in their radioprotection, as endogenous ROS production is suppressed while the intracellular GSH level becomes increased, thus conferring an antioxidant defense (see [110,111] and Figure 5). The Warburg effect also drives the dedifferentiation of hypoxic cancer cells by modulating chromatin accessibility via an acetyl-coenzyme A (CoA)-dependent mechanism and histone acetylation [112].

One of the consequences of the energy metabolism reprogramming in hypoxic tumor cells is acidosis through the overproduction of lactic and carbonic acids as a result of enhanced glycolytic and carbonic anhydrase activities. The hypoxia-induced acidosis development depends on HIF-1α and can stimulate malignant growth, as well as the resistance of hypoxic tumors to chemotherapy and radiotherapy [34,113,114,115]. Moreover, hypoxia-provoked acidosis can promote the EMT and maintain the cancer stemness [116], thus increasing the pool of radioresistant CSC-like cells in hypoxic tumors (Figure 5). Nevertheless, hypoxia-associated acidosis provides an additional option for the selective targeting of hypoxic tumors with radiosensitizers activated by low pH [117,118].

The major players in intratumoral hypoxia-associated energy metabolism reprogramming are hexokinase 2 (HK2), glucose transporter 1 (GLUT1) and pyruvate dehydrogenase kinase 1 (PDK1); their HIF-1-mediated expression ensures the Warburg effect and thereby contributes to the radioresistance of hypoxic tumors by enhancing the antioxidant capacity of the involved cancer cells [110,111,119]. In addition to its important role in the Warburg effect, HK2 can directly interact with HIF-1α, and this mortalin-mediated interaction, occurring at the outer membranes of mitochondria, modulates the voltage-dependent anion-selective channel 1 (VDAC1) activity to preserve hypoxia-stressed cancer cells from apoptosis [40]. As a result of hypoxia-induced energy metabolism reprogramming, the augmented PDK1 expression in hypoxic tumors can drive the PI3K/Akt/mTOR signaling that promotes the EMT and formation of the radioresistant CSC-like phenotype, as it was shown for PDK1-overexpressing hepatocellular carcinoma [120]. Another glycolytic enzyme, fructose-bisphosphate aldolase A, was shown to be induced by hypoxia in colorectal cancer cells and correlated to their chemo- and radioresistance and, also, to a poor prognosis [121].

The pentose phosphate pathway (PPP) is also involved in the hypoxia-induced rearrangement of energy metabolism in cancer cells, which yields them an increased GSH level and enhanced radioresistance ([56]; Figure 5). Using a knockdown technique with small hairpin (sh) RNA, Heller et al. [122] showed that transketolase-like protein 1—a key enzyme of PPP in some tumors and a target of HIF-1α—preserves hypoxia-stressed LNT229 glioma cells from the increase in the intracellular ROS level and is necessary for their adaptation to oxygen deficiency and the acquisition of radioresistance. In high-grade gliomas, the fluctuations in energy metabolism and radioresistance of the cancer cells appear to obey the circadian rhythm [57].

Although HIF-1α is thought to be the major regulator of energy metabolism reprogramming in hypoxic tumor cells [114,115], microRNAs and circular RNAs can also participate in this regulation. Therefore, miR-21 expressed in non-small cell lung cancer cells increases their radioresistance via the upregulation of HIF-1α-promoted glycolysis [123]. In hepatocellular carcinoma, hypoxia was shown to downregulate the miR-125 expression, which is accompanied by the expression of HK2 and enhancement of glycolysis [124]. Besides, there is the hypoxia-responsive mutual regulation of MiR-199a-5p and HIF-1α in hepatocellular carcinoma cells: under hypoxia, the miR-199a-5p expression is suppressed by upregulated HIF-1α, which leads to the stimulation of glycolysis and realization of the Warburg effect, whereas, under normoxia, MiR-199a-5p directly targets the 3’-untranslated region (UTR) of HIF-1α transcripts, thereby suppressing both the expression of HIF-1α downstream glycolysis-related genes, which reduces the glucose uptake and lactate production [27]. It was found in a model with breast cancer cells that circular RNA circRNF20 promotes the Warburg effect by harboring miR-487a and, thus, allows the HIF-1a-dependent expression of HK2 via the circRNF20/miR-487a/HIF-1α/HK2 axis [125]. In another model with breast cancer cells, circle RNA circABCB10 was shown to contribute to their radioresistance by upregulating glycolysis via the miR223-3p/profilin-2 regulatory axis [126]. Some tumor-suppressor microRNAs may increase the radiosensitivity of hypoxic tumors by inhibiting glycolysis in them via the downregulation of HIF-1α, as it was shown for miR-33a in melanoma cells [127]. In contrast, onco-miR-365 expressed in cutaneous squamous cell carcinoma abolishes the Homeobox A9 (HOXA9)-mediated downregulation of HIF-1α, along with HK2, GLUT1 and PDK1, thereby enhancing glycolysis [128].

Adenosine monophosphate (AMP)-activated protein kinase (AMPK) is a cellular energy sensor adapting cells to energy starvation when the cytosolic ATP/ADP/AMP ratio decreases [129]. Actually, AMPK acts as a switcher of cancer cell metabolism toward a more energy-saving mode, with an increased uptake of glucose and repressed lipogenesis; AMPK activation is interrelated with the functioning of HIF-1 (reviewed in [130]). In human hepatoma cells, AMPK-activated histone deacetylase 5 (HDAC5) deacetylated HSP70, which, in turn, mediated the HSP90-HIF-1α interactions, followed by the nuclear translocation of HIF-1α and induction of the HIF-1-dependent transcription response [19]. The AMPK signaling pathway is known to be stimulated in hypoxia-stressed tumor cells, which may elevate their radioresistance (Figure 5). It was demonstrated in SV40-transformed human fibroblasts [131] and glioblastoma cells [132] that hypoxia-induced AMPK activation resulted in the enhancement of both the expression/activation of Ataxia telangiectasia-mutated (ATM) and the activation of DNA-dependent protein kinase catalytic subunit (DNA-PKCs)—two enzymes contributing to DNA double-stranded break repair. In glioblastomas, hypoxia-induced AMPK activation triggers autophagy [133,134], which can also enhance the tumor cell radioresistance (described in Section 6). Additionally, AMPK, being activated in cancer cells undergoing hypoxia and nutrient starvation, can promote the EMT [135], which has to result in the accumulation of radioresistant CSC-like cells and, therefore, to impair the hypoxic tumor responsivity to radiotherapy (see Section 7). An involvement of AMPK in the CSC adaptation to the conditions of oxygen/nutrient limitation within hypoxic niches was also reported [136,137]. Therefore, as well as HIF-1α, AMPK is implicated in the radioprotective mechanisms that act in hypoxic tumors (Figure 5).

Here, it should be mentioned that tumor necrosis factor receptor-associated protein 1 (TRAP1), a member of the HSP90 subfamily, also contributes to adapting the cancer cell energy metabolism to hypoxic conditions [138,139]. Being the intramitochondrial ATP-dependent chaperone, TRAP1 inhibits succinate dehydrogenase, the complex II of the respiratory chain, thus facilitating the switch to glycolysis (the Warburg effect) that occurs in malignant cells [140,141]. Besides, TRAP1 in CSCs (which are known to be radioresistant) assists them in adapting to the energy-unfavorable conditions of hypoxic niches and attenuating ROS production (reviewed in [142]). Although there are not yet publications directly linking TRAP1 and the elevated radioresistance of hypoxic tumors, such links seem highly likely, as TRAP1 activity may potentiate the antioxidant capacity of cancer cells adapted to hypoxia [138].

### 3.2. Targeting Cellular Energy Metabolism to Radiosensitize Hypoxic Tumors

Taking into consideration that it is HIF-1 that mediates the expression of HK2, GLUT1 and PDK1 and formation of the “Warburg phenotype” within hypoxic tumors [114,115], the inhibitory targeting of HIF-1α may abolish both the hypoxia-associated energy metabolism reprogramming in tumor cells and their high radioresistance conferred by this reprogramming (see the previous subsections and Table 1). By another way, it seems possible to target certain components of cellular energy metabolism to sensitize hypoxia-adapted cancer cells to radiotherapy. Table 2 presents a list of small molecule inhibitors of the energy metabolism that, after the relevant model studies, have been suggested as potential radiosensitizers of hypoxic tumors. 

Taken together, the data of Table 2 prove the rationale of the inhibitory targeting energy metabolism of hypoxia-stressed cancer cells to radiosensitize hypoxic tumors. Although the Warburg effect and avid uptake of glucose are extremely important for the adaptive and radioprotective mechanisms in tumor cells undergoing hypoxia, several mitochondrial targets also seem significant for the radiosensitization of these cells [75,147,148,149]. The mitochondrial metabolism-targeting antiparasitic drugs atovaquone, ivermectin, proguanil, mefloquine and quinacrine were suggested as potentially effective radiosensitizers for hypoxic high-grade gliomas [149].

Apparently, pretreatments with suitable inhibitors of the glucose transport/metabolism or blockers of certain mitochondrial reactions would improve the therapeutic effect of irradiation on tumor cells rendered more radioresistant through their adaptation to hypoxia. This approach might be useful for both external irradiation and immunoradiotherapy and radioembolization. The problem, though, is that, at the present time, such inhibitors are still not approved for combining with radiotherapy to better target hypoxic tumors. After trials, 2-ME2 and dichloroacetate, despite their promising effects obtained in model studies [89,119], have not been permitted for cancer treatment. The two antidiabetic agents, metformin and phenformin, fairly well radiosensitized colorectal cancer cell lines, being more radioresistant under hypoxia; this phenomenon was suggested to be due to the mitochondrial complex I inhibition by the drugs [148]. In another study, metformin was shown to overcome the hypoxia-associated radioresistance of human lung cancer xenografts [150], although the achieved radiosensitizing effects were there explained by drug-induced oxygenation. The Phase I dose-finding study of metformin in combination with cisplatin and radiotherapy yielded encouraging results for patients with head and neck squamous cell cancer [151], so that, in a delayed perspective, metformin may be adopted as a radiosensitizer for certain types of hypoxic tumors.

Meanwhile, the situation with AMPK needs to be clarified. On one hand, AMPK seems to be a promising molecular target for radiosensitizing hypoxic tumors, because the downregulation of AMPK may increase the cell radiosensitivity, as it was shown by some authors [131,132]. On the other hand, the radiosensitization of hypoxic tumor cells with metformin was accompanied by the AMPK activation in them [148,152]. The development and characterization of novel AMPK inhibitors are presently continued [153], and all the newly discovered ones are to be tested in the relevant model studies aimed at sensitizing hypoxic tumors to radiotherapy.

As for TRAP1, this intramitochondrial chaperone has been characterized as a feasible target for future cancer treatments, while selective inhibitors of TRAP1 are presently developed and tested in cancer-related models [154]. Some of those inhibitors may be applicable for sensitizing hypoxic tumors to radiotherapy, but this option needs to be examined.

It should be added that, as well as it was suggested for the HIF-1 inhibitors (see Section 2.2), agents inhibiting the glucose uptake, energy metabolism or AMPK activation may aggravate the pathogenesis of some concomitant diseases (i.e., ischemia or diabetes); therefore, the trials and administration of such agents must be performed with great caution.

## 4. HSF1-Mediated Heat Stress Response and Heat Shock Proteins (HSPs)

The so-called heat shock proteins (HSPs) are molecular chaperones that regulate the folding of polypeptide chains and assist the stressed cell to renature or degrade stress-damaged proteins [155]. The major HSPs (HSP90, HSP70 and HSP27) are expressed constitutively, while their inducible forms are products of the transcriptional “heat stress response” that is mediated by a heat shock transcription factor 1 (HSF1) [156]. In vivo, HSF1 activation and HSP induction may locally occur upon ischemia, acidosis, inflammation, edema and some other tissue insults. Stress-induced intracellular HSP accumulation contributes to post-stress cell survival/recovery. Both HSF1 and HSPs are implicated in oncogenesis and responsible for the tumor resistance to chemo- and radiotherapy [157,158]. HSF1 and HSPs are directly involved in the tumor cell response to hypoxia.

### 4.1. Implication of HSF1 in the Cancer Cell Responses to Hypoxia and Radiation Exposure

Under nonstressful conditions, cytosolic HSF1 is in the complexes with HSP90 and HSP70, which preserve it from activation. In cells experiencing any proteotoxic stress (heating, hypoxia, energy starvation, acidosis or others), both HSP90 and HSP70 are recruited by stress-denatured protein molecules and, hence, liberate HSF1; the latter, becoming “free”, is activated via its phosphorylation/trimerization and translocated to the nucleus, where it binds to the “heat shock element” (HSE) in the promoter regions of the inducible *HSP* genes to initiate their transcription [156,157]. Besides the intracellular level of denatured proteins, certain protein kinases (stress kinases p38, JNK and ERK1/2) and protein phosphatases regulate the HSF1 activity in mammalian cells [156].

HSF1 was shown to be activated in hypoxia-stressed cancer cells [159,160]. Being the main player in triggering HSP expression, HSF1 also regulates the HIF-1α expression and tumor-driving HIF-1-HuR pathway, some protein kinase-based signaling pathways, autophagy, the energy metabolism and the redox potential, as well as the expression of certain microRNAs and long noncoding RNAs [160,161,162,163]; such activities allow HSF1 to contribute to many traits of tumor cells, including their ability to adapt to hypoxia and survive radiation exposure.

HSF1 can confer the tumor cell radioresistance by upregulating inducible HSP90, HSP70 and HSP27, which protect against post-radiation cell death and replicative senescence (see [164,165,166] and the next three subsections). The functioning of HSF1 was found to be required for post-radiation cell cycle (G2) arrest and double-stranded DNA break repair [167]. Being activated in the intratumoral hypoxic niches, HSF1 is one of the endogenous drivers of EMT that increases a subpopulation of radioresistant CSC-like cells ([142]; Figure 5). Furthermore, hypoxia-activated HSF1 may augment the expression of MDR1, a membrane transporter whose expression is under the control of HSF1 [168]; if so, the overexpressed MDR1 may pump some small molecule radiosensitizers out the hypoxia-adapted cancer cell. This phenomenon, when it is manifested, may impair the beneficial action of some radiosensitizers, including the ones generated from hypoxia-activated prodrugs, while the latter are thought to be a very promising tool for radiosensitizing hypoxic tumors (reviewed in [169,170]). Probably, the efficacy of small-molecule radiosensitizers may sometimes be enhanced by inhibiting HSF1 and/or MDR1.

It seems likely that drugs that inactivate HSF1 are able to radiosensitize hypoxic tumors. In this connection, naphthazarin and its derivative S64 were shown to radiosensitize breast cancer MCF-7 cells [171] or inhibit the DNA-binding activity of HSF1 and deplete GSH in hypoxic colon cancer cells [172]; these findings suggest a potential use of both agents for targeting hypoxic tumors. Yoon et al. [173] found that 2,4-bis(4-hydroxybenzyl)phenol isolated from *Gastrodia elata* radiosensitizes lung cancer NCI-H460 cells via the dephosphorylation and degradation of HSF1; this compound may similarly act toward hypoxic tumors.

Under normoxia, such known inhibitors of HSF1 activation as quercetin, KNK437 and NZ28 exert a radiosensitizing effect on cancer cells treated with inhibitors of the HSP90 activity [166,174]; however, it remains to be established whether these drugs are able to radiosensitize hypoxia-adapted cancer cells as well. KNK437 was demonstrated to act as a radiosensitizer toward breast carcinoma cells and glioblastoma cells undergoing hypoxia [92], but the researchers indicated an HSF1-independent mechanism of the radiosensitization. Although those findings [166,171,172,173,174] suggest a possibility of the use of HSF1 inhibitors to radiosensitize hypoxic tumors, none of those agents can be applied in clinic. Meanwhile, various small-molecule inhibitors of HSF1 are currently being developed and tested in preclinical trials and considered as potential tools in the fight against cancer [175,176].

On one hand, there is a great need for clinically applicable inhibitors of HSF1 activation, which would enable sensitizing hypoxic tumors to radiotherapy. On the other hand, in the clinical setting, these inhibitors should be used with great care, as they may increase the sensitivity of patient’s tissues to chemotherapy and some pathophysiological states such as ischemia/reperfusion, inflammation, endotoxemia, etc. Probably, some additional modalities should be proposed in order to restrict the cell-sensitizing action of HSF1 inhibitors within the volume of target tumors.

### 4.2. HSP90 as a Potentially Druggable Target for Radiosensitizing Tumors

Being a member of the HSPC chaperone subfamily, cytosolic HSP90 possesses ATPase activity and interacts with its client proteins in an ATP-dependent manner [177]. Among the client proteins of HSP90, there are receptors of growth factors and steroid hormones; transcriptional factors, including HIF-1α, HSF1 and NF-κB; protein kinases and products of some oncogenes, etc. [158,177]. HSP90 participates in the regulation of the stability/activity of HIF-1α and its import to the nucleus, which largely defines the cancer cell adaptation to hypoxic stress [19,21]. Many of the HSP90 client proteins are regulators of cancer-related signaling and gene transcription, which ensure the unlimited and metastatic growth of malignant tumors, as well as the tumor resistance to chemo- and radiotherapy. If HSP90 is inhibited, these client proteins become inactivated, ubiquitinated and then degrade at proteasomes, which may lead to regressing or sensitizing tumors. Therefore, the inhibitors of HSP90 chaperone activity are considered as potential anticancer drugs, and various HSP90-targeting agents are currently being examined in preclinical and clinical trials (reviewed in [158,177,178]).

HSP90 is known to be implicated in the mechanisms of tumor radioresistance, as some of its client proteins are required for DNA damage repair and avoiding post-radiation cell death or senescence [158,177]. Moreover, the members of the HSP90 subfamily (including TRAP1) promote the EMT in intratumoral hypoxic niches, thereby increasing a number of the radioresistant CSC-like cells within the target tumor [142]. The TRAP1-mediated modulation of mitochondrial functioning in cancer cells may also be conducive to the elevated radioresistance of hypoxic tumors (see Section 3.1). Based on this, one can suggest that certain inhibitors of HSP90 activity are able to sensitize hypoxic tumors to radiotherapy.

According to Kim et al. [21], lung cancer cell lines became radiosensitized by HSP90 ATPase inhibitors, 17AAG or deguelin, via blocking the HSP90-HIF-1α interactions that are necessary for cancer cell radioresistance. In contrast, it was found on the other lung cancer cell line (H1339) that HSP90 activity inhibition with 17AAG or NVP-AUY922 conferred radiosensitization under both normoxic and hypoxic conditions via an HIF-1α-independent mechanism [179]. Likewise, inhibiting HSP90 activity with NVP-AUY922 was shown to sensitize hypoxic cells of the glioblastoma SNB219 and lung carcinoma A549 cell lines to radiation exposure; the radiosensitizing effects were associated with a slowing down of double-stranded DNA break repair and the reduced expression of antiapoptotic proteins Akt and Raf-1 [180]. The HSP90 inhibitors NVP-AUY922 and NVP-BEP800 suppressed the migration and invasion of both normoxic and hypoxic cancer cells (lung A549 carcinoma and glioblastoma SNB219 lines) subjected to irradiation [181].

Importantly, while carbon ion beam exposure, as a kind of high-LET radiation, is thought to be an effective modality to target hypoxic tumors and attenuate the oxygen effect (Figure 3; [4,10,11]), tumoral HSP90 appears to enhance cancer cell resistance to carbon ion irradiation [182]. In a model with murine osteosarcoma line LM8 and normal human fibroblast line AG01522, Li et al. [182] reported that PU-H71, an inhibitor of HSP90 activity, sensitized the cancer cells to carbon ion irradiation with only a slight sensitizing effect toward the normal fibroblasts. At the molecular level, HSP90 inhibition in PU-H71-treated osteosarcoma cells reduced the protein expression levels of Rad51 and Ku70, which are required for the homologous recombination pathway and the nonhomologous end-joining pathway, respectively, in the machinery of double-stranded DNA break repair [182]. A similar study was performed on cells of the human lung carcinoma A549 and H1299 lines, cervical carcinoma-derived HeLa-SQ5 line and human lung fibroblast HFL-III line [183]. In that comparative study, HSP90 inhibition with PU-H71 conferred the sensitization of cells of all the three carcinoma lines to both carbon ions and X-rays, whereas the effect of PU-H71 on the radiation response of normal fibroblasts was insignificant. As well as in the case of osteosarcoma cells [182], PU-H71-treated carcinoma cells exhibited a failure in double-stranded DNA break repair following carbon ion irradiation [183]. Thus, cell-permeable inhibitors of HSP90 activity may enhance the therapeutic effect of carbon ion beams on hypoxic tumors with minimal radiation damages to the surrounding normal tissues.

Many compounds that selectively inhibit HSP90 ATPase (including 17AAG, NVP-AUY922, NVP-BEP800 and others), despite the encouraging results demonstrated in preclinical trials, have not been approved for therapeutic use. In addition to the expectable problems with high toxicity and poor bioavailability of most of the HSP90-binding agents, there is another one connected with HSF1 activation resulting from HSP90 dysfunction [165,166,174]. Indeed, many of the HSP90 activity inhibitors stimulate the HSF1 activation-mediated induction of HSP70, HSP27 and MDR1, which can elevate the radioprotective and adaptive potential of the tumor cells that survived after the inhibitory treatment. This problem may be solved by combining the HSP90 activity inhibitors with inhibitors of HSF1 activation, as was proposed for enhancing the radiosensitization of cancer cells [166,174].

It should be noted that the chaperone function of HSP90 can be downregulated or upregulated by its acetylation/deacetylation [177,178]. Accordingly, LBH589 (Panobinostat), an inhibitor of histone deacetylase 6 (HDAC6), was used to inhibit the HSP90 activity in cells of several carcinoma cell lines that became radiosensitized in such a way [184]. Panobinostat is a permitted anticancer drug, and, although its administration can also exert therapeutic effects nonrelated to HSP90 inactivation, it seems interesting to examine this inhibitor (and other HDAC inhibitors affecting HSP90) in models with irradiated hypoxic tumors.

### 4.3. Roles of HSP70: A Radioprotector of Cancer Cells and Potential Target for Radiosensitizing Them

Members of the HSP70 (HSPA) subfamily are chaperones possessing ATPase activity that regulate the maturation, transport and degradation of protein molecules [185,186]. HSP70 works in cooperation with other chaperones such as HSP110, HSP90, HSP60, HSP40, HSP27 and others that are implicated in protein folding/degradation in the cytoplasm and organelles [185,186,187].

HSP70 induction in cancer cells being subjected to hypoxia or ischemia-mimicking stresses was established many years ago [188,189,190]. Later, it was shown that arterial embolization in rabbit liver VX2 tumors yields HSP70 overexpression [191]. Apparently, all those examples reflect typical stress-responsive mechanisms aimed at the tumor cell adaptation to hypoxic conditions. HSP70 is directly involved in the cancer cell response to hypoxia, as its chaperone activity is required for the folding, stabilization and nuclear translocation of HIF-1α [19,192]. HIF-1α degradation may be HSP70-dependent as well [22,193]. In turn, HIF-1α upregulates the HSPA2/HSP70-2 expression in cancer cells [190,194] that is important for their growth and survival. The HSP70 contribution to the EMT and generation of CSC-like cells is also significant for both the pathogenesis of cancer and the radioresistance of hypoxic tumors (reviewed in [142]).

In many studies, the intracellular HSP70 level in cancer cells was found to be correlated with their radioresistance [195,196,197,198,199,200]. The radioprotective function of inducible HSP70 was also revealed in cancer cells treated with HSP90 activity inhibitors [166,174]. Mechanistically, the contribution of intracellular HSP70 to cancer cell radioresistance may include the HSP70-mediated modulation of protein kinase activities [195] and cell cycle arrest [174], protection against apoptosis [166,196,197], implications in double-stranded DNA break repair [198] and, probably, other cytoprotective pathways. At the expression level, intracellular HSP70 can be upregulated by HSF1 [157,163], HIF-1α [190,194], Redd1 [199] and long noncoding RNA HOTAIR [200]. The ATP-dependent chaperone activity of HSP70 is known to be regulated via acetylation/deacetylation of the HSP70 molecule [19,201] and, also, via interactions of HSP70 with its cochaperones and cofactors such as HSP40, Hip, Hop, CHIP, Bag-1 and others [186].

Treatments with agents inhibiting the expression/activity of HSP70 are considered as feasible ways for better targeting malignancies [158,185,186]; probably, analogous treatments would be effectual for radiosensitizing hypoxic tumors as well. As the radiosensitizing effects of HSP70 knockdown were observed in various cancer cell lines [196,197,198,199], certain oligonucleotide vectors based on miRNAs or small interference (si) RNAs or sh RNAs or antisense DNA and able to downregulate the intracellular HSP70 level may be used for the so-called “gene therapy” aimed at the fight against hypoxic tumors. In a model with normoxic conditions, rituximab (a chimeric anti-CD20 monoclonal antibody that can be used in immunotherapy) conferred the radiosensitization of non-Hodgkin’s lymphoma cells that was associated with the decrease in intracellular HSP70 and enhanced post-radiation apoptosis [202]. Notably, a combination of ^131^I-rituximab and atorvastatin (an inhibitor of HIF-1α) demonstrated the encouraging results in the radioimmunotherapy of Burkitt’s lymphoma modeled in murine Raji xenografts [71]. It would be interesting to examine such a double-target (HSP70 + HIF-1α) approach on the radiation response of hypoxic lymphomas. Besides, small-molecule compounds are known that inhibit the HSP70-mediated maintenance of cancer stemness and generation of CSC-like cells through the EMT [142]; in principle, such compounds may also impair the radioresistance of hypoxic tumors.

Several cell-permeable inhibitors of the HSP70 chaperone activity have been characterized (reviewed in [155,185,186]); some of them exert antitumor effects in model studies and are considered as potential tools for chemotherapy. So far, there were no reports that any inhibitor of the HSP70 chaperone activity increased the radiosensitivity of cancer cells; however, it seems likely that, in the future, effective radiosensitizers of tumors (and/or hypoxic tumors) will be created on the basis of such inhibitors disrupting HSP70-dependent radioprotective mechanisms in cancer cells.

Intriguingly, in contrast to intracellular HSP70 protecting cancer cells, extracellular or cell membrane-bound HSP70 may kill or radiosensitize them [203,204]. Schilling et al. [203] found that the binding of extracellular HSP70 to phosphatidylserines at the cell surface causes the killing of both normoxic and hypoxic cancer cells. After comparative studying of several cancer cell lines, it was shown that the level of membrane-bound HSP70 is positively correlated with the cell radiosensitivity [204]. The described radiosensitization of p53 wild-type-expressing colon cancer (HCT116 line) and lung cancer (A549 line) cells with PXN727, a MDM2 inhibitor, was associated with the decrease in membrane-bound HSP70 [205]. Thus, the use of exogenous HSP70 or a modality somehow increasing the level of cell membrane-bound HSP70 in cancer cells may radiosensitize target tumors, including the hypoxic ones. Moreover, antibodies targeting HSP70 on the surface of cancer cells may be conjugated to radiosensitizer-containing microcarriers (e.g., liposomes, dendrimers or nanoparticles) for their delivery to the target tumor. Such an approach was used for the radiosensitization of glioblastoma cells in an in vitro model by means of serum albumin nanoparticles conjugated to the monoclonal cmHsp70.1 antibody and containing miRNA for the knockdown of survivin: the antibody ensured the specific targeting of the nanoparticles to the cancer cells, while miRNA-mediated survivin downregulation conferred the enhancement of post-radiation apoptosis [206]. Later, the same cmHsp70.1 antibody was conjugated to superparamagnetic iron oxide nanoparticles for their specific targeting to irradiated gliomas in vivo [207].

Thus, intracellular HSP70 and extracellular (membrane-bound) HSP70 may provide two absolutely different options for targeting and radiosensitizing hypoxic tumors.

### 4.4. HSP27: Targeting the “Small” Chaperone to Radiosensitize Tumors

This “small” HSP belongs to the HSPB subfamily of ATP-independent chaperones that can assist the protein refolding/degradation machinery and, also, protect the stressed cell from protein aggregation and apoptosis [158,208,209]. HSP27 undergoes phosphorylation in the p38/MAPK pathway, so that the oligomeric state and cytoprotective activities of HSP27 are regulated via its phosphorylation/dephosphorylation [208]. Being implicated in cancer cell signal transduction, HSP27 can influence the activation of protein kinase C (PKC), Akt, NF-κB and some other signaling pathways [208,209,210]. Besides, excess HSP27 promotes intracellular GSH accumulation, being a potent cytoprotectant from oxidative stress [208] and inhibitor of HSP90 activity [211].

A significant role of HSP27 in oncogenesis, cancer cell stemness and tumor cell resistance to therapeutics has been established [142,158,209]. There are data that HSP27 expression is increased in hypoxia-stressed cancer cells [212], while HSP27 can confer a tolerance to hypoxia in cancer cells [210] and CSCs [142,213]. It was shown that HSP27 interacts with HIF-1α [210], while HIF-1α and HSF1 regulate HSP27 expression in cancer cells [214]. All this characterizes HSP27 as one of the key players in the tumor response to hypoxia.

Importantly, intratumoral HSP27 acts as an endogenous radioprotector: silencing the *HSP27* gene in irradiated cancer cells of various origins increases their apoptosis and decreases their post-radiation survival [215,216,217,218,219,220]. Using HSP27 knockdown techniques, Guttmann et al. [221] found that HSP27 is required for ATM-mediated DNA double-stranded break repair in irradiated head and neck cancer cells, while irradiated tumor xenografts with downregulated HSP27 expression exhibited slower growth in nude mice. HSP27 overexpressed in salivary adenoid cystic carcinomas was reported to be associated with the transforming growth factor-beta (TGF-β)-induced EMT, cancer cell stemness and radioresistance [222]. Additionally, the friend leukemia integration 1 (Fli-1)-mediated upregulation of HSPB1 (HSP27) in glioblastoma cells was correlated with their resistance to temozolomide and radiation [223]. All these data allow one to consider HSP27 as a potential molecular target for radiosensitizing hypoxic tumors.

At present, there are not yet publications demonstrating that inhibition of the expression or activity of HSP27 results in the desirable radiosensitization of hypoxia-adapted cancer cells. However, such an opportunity seems quite likely, as there are reports that the inhibitory targeting of HSP27 in malignant cells can elevate their radiosensitivity [224,225,226]. Korean researchers have shown that a heptapeptide of the protein kinase C delta (PKC-δ) catalytic V5 region binds to HSP27 in lung carcinoma NCI-H1299 cells, thereby impairing the HSP27-mediated tumor cell radioresistance in vitro [224] and in the xenografts growing in mice [225]. The same research group also observed in vitro and in vivo tumor cell radiosensitization as a result of treatments with zerumbone, a cytotoxic component isolated from *Zingiber zerumbet smith* that disturbs the normal HSP27 dimerization [226]. Those data provide a proof-of-principle that the selective inhibitors of certain functions of HSP27 can overcome the tumor cell radioresistance; perhaps, similar approaches would be applicable toward hypoxic tumors as well. Taking into consideration that both the oligomeric status and activities of HSP27 depend on its phosphorylation/dephosphorylation, it seems possible to suppress the HSP27-mediated radioprotection of cancer cells by affecting the activation of certain protein kinases and phosphatases (de)modifying the HSP27 molecules (e.g., p38 MAPK, MAPKAPK2 and PP2A [208,209,210]).

Alternatively, overcoming tumor cell radioresistance may be achieved via the downregulation of HSP27 expression. Such a way was demonstrated by Hadchity et al. [217], who used OGX-427, a second-generation antisense oligonucleotide, to reduce the HSP27 level in radioresistant head and neck squamous cell carcinoma SQ20B cells. Being pretreated with OGX-427, the irradiated cancer cells exhibited HSP27 downregulation, along with enhanced apoptotic death and decreased clonogenicity [217]. OGX-427 (apatorsen) is currently being tested in Phase II clinical combinatorial trials so that the application of OGX-427 for radiosensitizing hypoxic tumors is not excluded in the future. Several small-molecule compounds have been described that are able to suppress HSP27 expression in CSCs [142]; theoretically, such agents may prevent the generation of radioresistant CSC-like cells in hypoxic niches. Overall, in regard to the radiosensitization of tumors, HSP27 seems to be a promising target for both inhibitors of HSP27 functional activity and suppressors of HSP27 expression. One of the problems is how to deliver such HSP27-targeting agents inside hypoxia-adapted cancer cells prior to radiotherapeutic procedures.

It should be added herein that, besides HSP90, HSP70 and HSP27, other HSPs may somehow be involved in the tumor response to hypoxia and also contribute to the radioresistance of hypoxia-adapted cancer cells. The implication of HSP105 in hypoxia-induced intracellular HIF-1α accumulation and activation has been suggested [227]; whether HSP105 (or HSP60, HSP40, HSP32 and others) can be a target for the radiosensitization of hypoxic tumors remains to be examined.

### 4.5. HSF1, HSPs and the Radiosensitizing Effects of Hyperthermia

Local hyperthermia may be used as a clinically applicable method for radiosensitizing hypoxic tumors (see [4,228] for a review), especially since hypoxic regions within malignancies can be detected in vivo by means of special techniques (e.g., positron-emission tomography). It has been shown in several studies [229,230,231] that hyperthermic treatments can radiosensitize hypoxic tumors by reoxygenating them. At the physiological level, this oxygenation is largely due to the vasodilating action of hyperthermia that increases the blood flow across tumor tissues. Accordingly, the increased oxygen concentration in such heated tumors can enhance their radiation response and yield therapeutic benefits. It should be noted that, besides its beneficial effects, the hyperthermia-enhanced perfusion of hypoxic tumors with blood may have negative consequences, such as the stimulation of tumor growth and increase in the numbers of circulating tumor cells and metastases.

At the cellular and molecular levels, the heat-induced radiosensitization of cancer cells is realized via multifactor mechanisms and pathways that are accompanied by changes in the cell cycle and signaling, DNA damage response, HSP expression, etc. (reviewed in [232,233]); all this impairs the radioprotective potential of cancer cells and enhances their post-radiation death in heat-treated tumors. Taking into consideration that HSF1 and inducible HSPs are some of the major players in cancer cell responses to both hyperthermia [232,233] and hypoxia (see Section 4.1, Section 4.2, Section 4.3, Section 4.4, above), one can suggest that the inhibitory targeting of the HSF1 activation/HSP induction pathway and/or the functional activities of major HSPs (HSP90, HSP70 and HSP27) would improve the radiosensitizing effects of hyperthermia toward hypoxic tumors. Moreover, as HSF1 and all the major HSPs are known to promote the EMT and active migration of CSC-like cells [142], inhibiting those HSF1- and HSP-mediated mechanisms may decrease the risk of metastasis dissemination due to hyperthermia-enhanced blood circulation across target tumors. Such speculations have also motivated the development of clinically applicable inhibitors of HSF1 and HSPs.

## 5. Endoplasmic Reticulum Stress and Glucose-Regulated Proteins (GRPs)

Endoplasmic reticulum (ER) stress is induced by hypoxia, hypoglycemia, ion imbalance or some other exposures that compromise protein folding within the ER, thereby triggering the “unfolded protein response” (UPR) and expression of glucose-regulated proteins (GRPs) [234,235]. Being molecular chaperones, the newly expressed GRPs help the stressed cell to fulfil the “protein quality control” by catalyzing the renaturation or degradation of damaged proteins in the ER and mitochondria. Among inducible products of the UPR may also be apoptosis-promoting ones, such as the C/EBP-homologous protein (CHOP) and caspases that alleviate the post-stress elimination of severely suffered cells. Besides their implication in chaperoning, GRPs are regulators of signaling, Ca^2+^ homeostasis, apoptosis, autophagy, protein transport and secretion, the immune response, etc. [234,235]. As GRPs in cancer cells are conducive to malignant growth, the EMT and maintenance of cancer cell stemness and tumor chemo- and radioresistance (see Figure 5), some GRP-targeting agents are thought to be potentially applicable for repressing and sensitizing malignancies [142,236].

### 5.1. ER Stress and Radioresistance of Hypoxic Tumors

Hypoxia-induced ER stress and the UPR in cancer cells may lead to opposite outcomes: either cell survival and adaptation or cell death [234,235,236]. Therefore, ER stress may diversely affect the radiation response of cancer cells undergoing hypoxia. For example, ER stress with proapoptotic CHOP overexpression caused by insulin-like growth factor stimulation was shown to confer the radiosensitization of hypoxic cells of the pancreatic cancer cell line AsPC-1 [237]. Similar to that, celecoxib (an inhibitor of cyclooxygenase-2) conferred the radiosensitization of hypoxic cells of glioblastoma cell lines through the drug-provoked ER stress in them [238].

On the contrary, the ER stress-associated double-stranded RNA-activated protein kinase (PKR)-like endoplasmic reticulum kinase (PERK)/eukaryotic initiation factor 2α (eIF2α)-dependent arm of the UPR induces the uptake of cysteine and glutathione synthesis in hypoxic tumor cells (U373-MG and HCT116 cell lines), thus ensuring their protection against ROS and ionizing radiation [239]. In an in vitro breast cancer model, lysosome-associated membrane protein 3 (LAMP3) induced by the PERK/ATF4 arm of the UPR during hypoxia was shown to be required for DNA double-stranded break repair in cancer MDA-MB-231 cells and their radioresistance [240]; both LAMP3 knockdown and the chemical inhibition of PERK with GSK2606414 exerted radiosensitizing effects.

Probably, cell-permeable modulators of ER stress shifting the UPR in hypoxic cancer cells toward cell death would act as radiosensitizers of hypoxic tumors. Some beneficial ER stressors may be identified among the known drugs [241] or designed as novel compounds with the assistance of in silico modeling. At least, celecoxib seems quite suitable for the radiosensitization of hypoxic glioblastomas [238]; all the more, this anti-inflammation drug is able to increase the radiosensitivity of radioresistant CD133+ glioblastoma CSC-like cells [242]. In model systems with normoxic conditions, celecoxib also manifested the properties of a radiosensitizer toward human FaDu squamous cell carcinoma cells [243] and non-small cell lung cancer cells [244]. All of this should be a reason for radiotherapists to test celecoxib in trials for the radiosensitization of hypoxic tumors.

### 5.2. GRPs as Potential Targets for Radiosensitizing Hypoxic Tumors

GRP78 (also named as BiP or HSPA5) is largely localized to the ER, where this ATP-consuming chaperone functions as a master regulator of the ER stress-associated UPR [234,235,236]. GRP78 is induced in tumor cells stressed by hypoxia/hypoglycemia and plays an important role in their adaptation to stressful conditions and, also, in their chemoresistance [236]. The cytoprotective and antiapoptotic activities of overexpressed GRP78 appear to potentiate the radioresistance of hypoxia-adapted cancer cells, so that the artificial downregulation of GRP78 should be considered as a potential way to radiosensitize hypoxic tumors. In support of that, the specific cleavage of GRP78 in glioblastoma cells (U251 line) treated with the fusion protein epidermal growth factor (EGF)-SubA increased the cancer cell sensitivity to both temozolomide and ionizing radiation [245]. The question is, though, whether the EGF-SubA-based approach is applicable in vivo against hypoxic tumors.

GRP75 (known as mortalin or HSPA9) has its main location in the mitochondria and exhibits antiapoptotic potential; this chaperone is thought to be one of the factors defining tumor growth/resistance to therapeutics [234,235]. GRP75 participates in regulation of the mitochondrial functions and cancer cell adaptation to hypoxic conditions. It was revealed in cells of hepatocarcinoma (Huh7 line) and cervical carcinoma (HeLa line) that, under hypoxia, mortalin (GRP75) mediates both the translocation of non-phosphorylated HIF-1α to the outer mitochondrial membrane and its association with VDAC1 and HK2, which leads to the modulation of VDAC1 activity and, as a result, the protection of the hypoxia-stressed cancer cells from apoptosis [40]. As the same involvement of HIF-1α in cancer cell protection from apoptosis took place in the case of genotoxic treatments with etoposide or doxorubicin [40], the analogous GRP75-dependent antiapoptotic mechanism may protect hypoxia-adapted cells from apoptosis following radiation exposure.

Importantly, both GRP78 and GRP75 promote the hypoxia-induced EMT and generation of radioresistant CSC-like cells in hypoxic niches; some approaches and agents used in model studies to inhibit the contribution of all major GRPs to the formation/maintenance of a CSC phenotype were considered in a recent review [142]. It should be noted that some part of GRP78 is expressed on the surface of CSCs [142,246]. On one hand, this phenomenon is conducive to the cancer cell stemness that appears to potentiate the radioresistance of hypoxic tumors. On the other hand, GRP78 exposed on the CSC surface was characterized as a unique outer marker of CSCs that turns this chaperone into a potential target for selective attacking CSCs with specific anti-GRP78 antibodies conjugated to toxins or radionuclides (discussed in [142,246]).

Thus, some agents modulating hypoxia-associated ER stress toward the induction of CHOP and apoptosis, as well as selective inhibitors of GRPs, should be considered and tested as potential radiosensitizers for hypoxic tumors.

## 6. Hypoxia-Responsive Autophagy

### 6.1. Implication of Autophagy in Cellular Homeostasis and Stress Response

A scenario of autophagy (or macroautophagy) is realized via the surrounding of some cytoplasmic components (e.g., mitochondria) by the newly formed double membrane (phagophore) to enclose them into a vesicle (autophagosome), which, afterwards, becomes fused with a lysosome, thus generating an autolysosome where the isolated contents undergo degradation and recycling [247]. Under normal conditions, such a catabolic process maintains cellular homeostasis/renewal while postponing cellular senescence. The activation of autophagy in stressed cells may yield two opposite outcomes: (1) cytoprotection, resulting in cell survival/recovery and the adaptation to stressful conditions, or (2) cell suicide via lethal self-eating, i.e., an autophagic death [241,242,243,244,245,246,247,248,249]. Autophagic cell death is one of the mechanisms of elimination of irradiated cancer cells [248,249].

Autophagy-related genes (Atg) and their products are known to be involved in autophagy machinery; such proteins as Beclin-1, Atg3, Atg5, Atg7, Atg12, LC3 and others participate in the phosphorylation-dependent regulation of the induction and execution of autophagy [247,248,249]. Besides, HIF-1α and components of signaling pathways such as c-Jun NH_2_-terminal kinase, PI3K, Akt, mTOR and AMPK (see [29,51,52,53,54,133,134]; Figure 4 and Figure 5) and, also, microRNAs and long noncoding RNAs (see [29,54] and Section 8) can be stress-responsive regulators of autophagy.

Autophagy in cancer cells is thought to contribute to both their malignant growth and their resistance to therapeutics (reviewed in [247,248,249]). Moreover, autophagy appears to be one of the cancer stemness-maintaining mechanisms that allows CSCs to renew themselves and evade apoptosis or senescence [250]. On the one hand, the inhibition of autophagy may repress tumors or sensitize them to chemotherapy and radiotherapy; on the other hand, some autophagy-inducing agents may beneficially stimulate autophagic death in treated tumor cells [247,248,249]. In any event, autophagy seems to be a significant target for anticancer therapy.

### 6.2. Autophagy and Radioresistance of Hypoxic Cancer Cells

Hypoxia-induced autophagy helps cancer cells adapt to metabolic stress resulting from the limitation of oxygen and nutrients [247,248,249]. Importantly, autophagy can affect the radiation response of hypoxia-stressed cancer cells (Figure 4 and Figure 5). It was shown in many studies that hypoxia-induced autophagy can contribute to the radioresistance of hepatoma [29], osteosarcoma [53], colon cancer [54], breast cancer [51,240,251,252], lung cancer [52,253,254], prostate cancer [255,256,257], bladder cancer [258] and glioma (glioblastoma) cells [29,259,260].

Meanwhile, the autophagy-based mechanisms of radioprotection in hypoxia-adapted cancer cells might vary and include the contribution of hypoxia-induced autophagy to double-stranded DNA break repair [240,251]; reducing the intracellular ROS level [53,254]; Parkin-mediated digesting mitochondria [252]; expressing proto-oncogene serine/threonine-protein kinase PIM-1 [249]; activating protein kinases Akt, mTOR and P70S6K [29,51] and c-Jun [52], involving HIF-1α [29,51,52,53,54] and/or microRNAs [54,255,256,257] or a long intergenic noncoding RNA (lincRNA)-p21 [29]. Such a variety in the involved mechanisms and factors implies the multifaceted implication of autophagy in cancer cell responses to hypoxia and irradiation, as well as the possibility of a multitarget approach to the radiosensitization of hypoxic tumors via activating autophagy in them.

However, there are reports that hypoxic cancer cells, being irradiated, may die via an autophagic death [248,249]; this means that, in some cases, the induced activation of autophagy in hypoxic tumors may beneficially increase their radiosensitivity. Indeed, the induction of autophagy was shown to confer the radiosensitization of hypoxic renal carcinoma cells [261]. The radiosensitization of hypoxic breast tumor cells (MDA-MB-231 and MCF-7 lines) with NVP-BEZ235, an inhibitor of the PI3K/mTOR pathway, was accompanied by the induction of autophagy [37], which is in some contradiction with other published observations in which autophagy contributed to the radioresistance of breast cancer cells [51,240,251,252]. The radiosensitizing action of NVP-BEZ235 was due to the suppression of both HIF-1α expression and Akt activation [37], so that those effects might overcome or revoke the effects of autophagy induction. A decrease in autophagy was correlated with the elevated radioresistance of hypoxic colorectal cancer cells [262] and cervical cancer cells in which autophagy was suppressed by MiR-21 via the Akt/mTOR signaling pathway [263] and dependent on the EGF receptor (EGFR) expression [264]. Notably, autophagy may oppositely affect the radiation response of hypoxic tumors with a similar localization, i.e., to weaken it in colon cancer [54] and enhance it in colorectal cancer [262]. Therefore, autophagy-based approaches to the radiosensitization of hypoxic tumors may be quite opposite (either the inhibition or activation of autophagy) for different cases.

A number of publications reported that the inhibition of autophagy radiosensitizes hypoxic cancer cells. For instance, a cell-penetrating fused protein construct, TAT-ODD-p53, which consists of a TAT domain of human immunodeficiency virus-1, oxygen-dependent degradation domain (ODD) of HIF-1α and wild-type p53 was shown to selectively accumulate in hypoxic breast cancer cells and render them more radiosensitive by inhibiting the autophagic digestion of the mitochondria (mitophagy) [252]. Bafilomycin A1 and chloroquine (both are small-molecule inhibitors of autophagy that prevent the fusion of autophagosomes with lysosomes) radiosensitized breast cancer cells of the MCF-7 and MDA-MB-231 lines by deteriorating their post-radiation DNA repair and clonogenicity [240]. Likewise, chloroquine acted as a good radiosensitizer for hypoxic cells of other tumors, including lung cancer [52], bladder cancer [258] and EGFRvIII-positive glioblastoma [260]. Importantly, chloroquine is currently being tested in clinical trials—in patients with EGFRvIII-expressing glioblastomas; this drug has demonstrated encouraging effects that support a continuation of clinical studies [260,265].

Lin et al. [258] improved the radiosensitizing action of chloroquine on bladder cancer cells by encapsulating the complex between chloroquine and MnO_2_ into nanoparticles (NPs) prepared on the basis of human serum albumin. Those NPs were pH-sensitive, and after a reaction with H^+^/H_2_O_2_, they generated O_2_ in conditions of hypoxia-associated acidosis, thus attenuating both hypoxia and acidosis. Besides, the NPs released chloroquine in acidic conditions, which ensured the drug uptake by cancer cells and autophagy inhibition—this all yielded a synergy in the NP-conferred radiosensitizing effects [258]. Such a nano approach seems very promising, because it simultaneously enables achieving (i) oxygen saturation within the hypoxic regions of the target tumor, (ii) the suppression of autophagy-based radioprotective mechanisms and (iii) the in vivo selectivity of radiosensitizing effects that are only manifested under acidic conditions (i.e., in hypoxic tumors where acidosis is a typical phenomenon).

Sometimes, the radiosensitization of cancer cells may be associated with the induction of autophagy in them. In particular, the small molecule compound STF-62247 was shown to induce autophagy in renal cell carcinoma cells that conferred their radiosensitization [261]. The above radiosensitizing action of NVP-BEZ235 on breast cancer cells [37], among other effects, induced autophagy, along with the retarded DNA repair from radiation damages. Both these publications provided the proof-of-principle that at least, in some cases, small-molecule inducers of autophagy can be exploited as radiosensitizers for better targeting hypoxic tumors.

Silver nanoparticles (AgNPs) were used by Liu et al. [259] to radiosensitize hypoxic glioma cells (U251 and C6 lines). The AgNP-conferred radiosensitization was associated with the induction of destructive autophagy and enhanced apoptosis in the treated hypoxic glioma cells [259].

Thus, autophagy in cancer cells represents a dual possibility for the radiosensitization of hypoxic tumors, with either inhibitory or stimulatory targeting of the autophagic machinery in cancer cells. Notably, NPs may successfully be used in both approaches [258,259]. It seems likely that pretreatments aimed at the respective modulation of autophagy in hypoxic cancer cells would improve the outcomes of radiotherapy and radioembolization.

## 7. Hypoxia-Induced Generation of Radioresistant CSC-Like Cells

The major feature of CSCs is their tumorigenicity based on endless mitotic division and self-renewal. Due to their tendency to migrate, CSCs are the drivers of tumor invasion and metastasis spread. Being more resistant to chemo- and radiotherapy than non-stem cancer cells, CSCs may survive after treatment and be a cause of cancer relapse [266,267]. Therefore, CSCs seem to be the most significant target for anticancer therapy.

CSCs have characteristic phenotypic markers, including specific cell surface antigens and upregulated expression/activities of certain enzymes and membrane transporters [142,266,267]. Oncogenes, epigenetic regulators, signaling pathways and transcriptional factors are known that contribute to the development and maintenance of cancer stemness. In particular, the Notch and Hedgehog signaling pathways promote the self-renewal of CSCs by driving the expression of NANOG, Slug, Twist, sex determining region Y-box transcription factor 2 (SOX2) and octamer-binding transcription factor 4 (OCT4). Wnt signaling, being hyperactivated in CSCs, was shown to upregulate the expression of CD44, CD133, LGR5, aldehyde dehydrogenase (ALDH) and membrane transporter ABCG2 (all are phenotypic markers of CSCs) [266,267]. Besides, the two oncogenes Multiple Copies in T-cell Malignancy 1 (MCT-1/MCTS1) and B lymphoma Mo-MLV insertion region 1 homolog (BMI1); HSF1 and molecular chaperones, including HSPs, GRPs and others; epigenetic regulator PR domain-containing protein M14 (PRDM14); microRNAs and long noncoding RNAs are involved in the formation/maintenance of a cancer stem phenotype (reviewed in [142,266,267].

The intratumoral pool of CSCs is maintained or increased via their asymmetric or symmetric division, respectively. Importantly, drug-resistant and radioresistant CSC-like cells can be generated as a result of the dedifferentiation of non-stem cancer cells in response to the action of some humoral and microenvironmental factors or therapeutic agents. The EMT can occur in carcinoma cells, which transforms a part of them into fibroblast-like malignant cells expressing mesenchymal markers instead of the epithelial ones with a potent capacity toward migration [8,9,268,269]. Such a process is extremely important for cancer pathogenesis, because the EMT-generated CSC-like cells are responsible for tumor invasion in adjacent organs, penetration into the vasculature and the dissemination of metastases (Figure 2).

In vivo, the EMT with generation of CSC-like cells may be stimulated by certain hormones, cytokines, growth factors, hyaluronic acid, nitric oxide and other bioactive molecules that come from the extracellular matrix, tumor vasculature, secreted vesicles (exosomes), stromal cells, etc. [266,267]. Besides, radiation exposure, some drugs and, also, hypoxia can be the inducers of EMT or similar phenotypic modulations toward cancer stemness (reviewed in [8,9,142,268,269]).

### 7.1. Hypoxia-Induced Formation of the Radioresistant CSC Phenotype and “Runaway” of Migrating CSC-Like Cells from Therapeutic Radiation Exposure

Among the established EMT-inducing stimuli, there are hypoxia, nutrient limitations and acidosis in tumor regions with failed blood circulation (Figure 4 and Figure 5). These hypoxia-associated stimuli activate the Hedgehog, Wnt, Notch and Cripto-1-mediated signaling pathways and, also, transcription-regulating factors HIF1α, HIF2α, STAT3, NANOG, OCT4, HSF1 and NF-κB, which ensure the initiation and fulfilment of the EMT program in hypoxic carcinoma cells [8,9,142,268,269]. Additionally, exosomes secreted by hypoxia-stressed tumor cells are particularly conducive to the induction of EMT and development of a CSC-like phenotype within hypoxic niches (see [270], Figure 2 and Section 9). MicroRNAs, long noncoding RNAs and other epigenetic regulators can also modulate the mechanisms of hypoxia-induced EMT and the generation of CSC-like cells [271,272,273,274].

The fact that hypoxia-induced EMT and cancer stemness development increase the radioresistance of involved tumor cells has been shown in several relevant models [64,275,276,277,278,279], although the radioprotective mechanisms can somewhat differ in various cases. For example, Wang et al. [64] found that CD133-positive (CSC-like) cells of human laryngeal squamous carcinoma (Hep-2) are more radioresistant, and their radioresistance is enhanced under hypoxia, which is associated with an increase in DNA-dependent protein kinase (DNA-PK) activity. Similarly, ALDH-1-positive (CSC-like) cells from the cervical cancer lines HeLa and SiHa exhibited a radioresistant phenotype that was enhanced with hypoxia and was due to the preferential activation of the DNA damage checkpoint response and an improvement to post-radiation DNA repair [279]. Twist1, one of the drivers of the EMT, was found to be elevated in hypoxia-treated cervical cancer (SiHa) cells, which increased their radioresistance, along with an increased nuclear EGFR localization and expression levels of nuclear DNA-PK [277]. In another case, transgelin 2 (an actin-binding protein), overexpressed in hypoxic non-small cell lung cancer cells, activated the insulin-like growth factor 1 receptor β (IGF1Rβ)/PI3K/Akt pathway via recruiting the focal adhesion kinase to the IGF1R signaling complex; this conferred the Snail1 stabilization and EMT-based selection of the cells resistant to γ-irradiation [278]. One of the factors defining the enhanced radioresistance of CSC-like cells is their lower proliferative activity versus non-stem cancer cells; being generated as a result of hypoxia-induced EMT, some CSC-like cells sojourn in a state of mitotic quiescence, which makes them more radioresistant [142,266,267]. The high activity of telomerase in CSCs [280] may help them to evade post-radiation replicative senescence. CSCs are known to have improved DNA damage response machinery that includes the expression of the Snail, ERCC1 and NBS1 proteins implicated in nuclear DNA repair after irradiation. Besides, dysregulation of p53-dependent apoptosis and the enhanced expression of antiapoptotic proteins survivin and Bcl-2 can contribute to the formation of radioresistant and apoptosis-resistant CSC phenotypes (reviewed in [142,266,267]. The metabolic reprogramming, low ROS production, intracellular accumulation of GSH and potent antioxidant capacity are intrinsic to CSCs and thought to be endogenous factors enhancing the radioprotective potential of CSCs [142,266,267]. The increased levels of HSPs, GRPs and other molecular chaperones in CSCs are also associated with the high radioresistance of these cells [142]. There are data that the elevated autophagic activity in CSCs promotes their radioresistance as well [281,282]. According to Wozny et al. [65], under hypoxia, CSCs may earlier accumulate HIF-1α compared to non-stem cancer cells; this may define the better readiness of HIF-1-mediated radioprotective mechanisms in CSCs residing in hypoxic niches. Other researchers suggested that the radiosensitivity of CSCs and CSC-like cells does not depend on the presence or absence of oxygen in their microenvironment [283]; if so, the pretreatment with the artificial oxygenation of hypoxic tumors may not ensure the complete success of radiotherapy.

The EMT-associated phenotypic modulation alters the gene profile expression, metabolism, composition and dynamics of the cytoskeleton, cell polarity, cell–cell contacts and cell–matrix interactions in such a way that it allows CSC-like cells to actively migrate within and from hypoxic niches (see [8,9,142,268,269]; Figure 2). Due to the specific alterations in the cytoskeleton and intercellular contacts, the expression of integrin β1 and expression/secretion of heparanase and extracellular proteinases such as matrix metalloproteinases (MMPs) and urokinase-type plasminogen activator (uPA), EMT-generated CSC-like cells are able to easily move across the extracellular matrix and invade nearby tissues and organs or penetrate into the blood and lymphatic vessels to spread metastases [142,266,267]. As the EMT induction and emergence of metastasis-forming CSC-like cells occur in hypoxic niches of aggressive malignancies, this may also impair the curative effects of radiotherapy. Indeed, under therapeutic modalities with long-term or serial radiation exposures (e.g., fractionated external irradiation, radioembolization, radionuclide therapy or brachytherapy), some EMT-generated CSC-like cells may survive, because they have the time to migrate and penetrate into the vasculature and then be transferred away by the flows of blood or lymph to a patient’s body regions outside the cancer cell-killing impact of photons or particle beams (see Figure 2). HIF-1 activity appears to be required for the translocation of irradiated CSC-like cells toward blood vessels [63]. Such direct “runaway” of CSC-like cells from the primary tumors undergoing therapeutic radiation exposure is evidently related to a high risk of the formation of distant metastases and poor patient outcomes.

Although the hypoxia-induced EMT program is realized in carcinomas (i.e., in malignancies derived from the epithelium), similar modulations of tumor cell phenotypes can also occur in hypoxic regions of sarcomas, gliomas, lymphomas and other types of solid tumors [8,9,268,269]. These EMT-resembling phenotypic modulations result in the appearance of less differentiated and more mobile tumorigenic (CSC-like) cells exhibiting a prominent tendency to migrate and form metastases. Such CSC-like cells in hypoxic niches of nonepithelial tumors may also evade the radiation exposure when they migrate and intrude on the vasculature and, then, are translocated outside the irradiation target zone owing to the circulation of blood and lymph (Figure 2). This phenomenon seems to be one of the causes of a low efficiency of radiotherapy toward hypoxic tumors.

### 7.2. Targeting CSCs and EMT to Overcome the Radioresistance of Hypoxic Tumors

Many research laboratories are currently continuing their efforts to develop therapeutic modalities for targeting CSCs [284,285]. In this respect, the EMT-generated CSC-like cells residing in hypoxic niches are also a significant target for anticancer therapy; in particular, the killing or radiosensitization of those cells would enable improvements patient outcomes under the irradiation of hypoxic tumors.

Being rather resistant to γ-photons and X-rays, CSCs are more sensitive to the cytotoxic action of neutrons [286], protons [287,288,289] and α-particles [290,291]. In contrast to photon radiation, which may induce the EMT and increase the fraction of CSC-like cells in target tumors [142,292], proton beams appear to reduce both the cell migration ability and amounts of CSC-like cells in irradiated tumor cell populations, as was demonstrated for non-small cell lung carcinoma A549 cells exposed to proton irradiation [288]. Therefore, irradiation with proton beams helps to minimize the risk of the metastatic scenario realization shown in Figure 2. An additional advantage of targeting CSCs with proton beams may be calreticulin expression on the CSC surface, as it was found in proton-irradiated breast CSCs; such a response provoked the cytolytic T-lymphocyte attack that killed the CSCs that survived after proton exposure [293]. It should be added that HIF-1 mediates the CD47 expression in breast cancer cells, thereby promoting their stemness and allowing them to evade phagocytosis [294]. Taking into consideration both references [293,294], one can suggest that combining proton irradiation with inhibitors of HIF-1 is to enhance the post-treatment immunogenic elimination of breast CSCs.

Besides proton beams, therapy with α-particles was also suggested as a curative modality to target prostate CSCs [290] and glioma stem cells [291]. In turn, carbon ion irradiation has been suggested as a promising approach to the eradication of CSCs [295] and/or suppression of their migration/invasiveness [296,297]. In studies with high-grade glioma, carbon ion irradiation was shown to be effective in the eradication of both hypoxic cancer cells and CSCs and, also, rendering the tumor xenografts more vulnerable to immune attacks [298]. All these data characterize the beams of protons or carbon ions and, also, α-particles as suitable therapeutic agents for targeting CSCs and CSC-like cells in hypoxic tumors. Indeed, such radiation exposures can effectively be used against hypoxic tumors, because they eliminate CSCs or at least impair their ability to migrate/invade. Nevertheless, CSCs appear to have certain internal resources that help them to resist radiation particle-based therapy, while some agents may be exploited to overwhelm the contribution of those resources. For example, UCN-01, a checkpoint kinase (Chk1) inhibitor, in combination with all-trans retinoic acid (ATRA), sensitized head and neck squamous cell carcinoma stem cells (SQ20B-CSCs) to both photons and carbon ions [299]. In addition, the radioprotective mechanisms mediated by HSP90 [182,183] and HIF-1α [65,297] may attenuate the cytotoxic effects of carbon ion beams on CSCs and hypoxia-adapted non-stem cancer cells, but the inhibition of HSP90 or HIF-1α enables the enhancement of those effects.

Notably, CSCs and CSC-like cells can be fairly sensitive to hyperthermia [300,301]. While local hyperthermia is thought to be a clinically applicable method for radiosensitizing hypoxic tumors [4,228], its effectiveness may partly be due to targeting radioresistant (but thermosensitive) CSCs.

Molecular chaperones (HSPs, GRPs and others) are known to play an important role in the viability of CSCs and manifestation of their tumorigenic traits [142]. Both intracellular and extracellular chaperones have been suggested as potential targets for suppressing or reversing EMT and killing or sensitizing CSCs in vivo [142]. It seems likely that some chaperone-targeting agents would be able to radiosensitize hypoxic tumors by reducing in them the population of radioresistant CSC-like cells, but this suggestion needs an experimental examination.

Of course, great expectations are associated with the search for pharmacological agents capable of targeting CSCs in vivo. Although a universal anti-CSC drug has not yet been developed, various natural and synthetic compounds are currently being tested in the relevant models with the aim to discover suitable agents for targeting the EMT and CSCs [284,285]. In this respect, the inducers of differentiation deserve attention, since some of them can reverse the EMT and transform CSCs into more differentiated (and more sensitive to therapeutics) non-stem cancer cells. The above agent ATRA has been shown to induce the differentiation and radiosensitization of SQ20B-CSCs [299] and breast CSC-like cells [302]; in both these references, the differentiating and radiosensitizing effects were due to the inhibition of ALDH activity by ATRA. The high radiosensitizing potential of ATRA toward CSCs was confirmed by in silico modeling [303]. 2-Methoxyestradiol (2-ME2) inhibited the HIF-1α expression and reversed the EMT in nasopharyngeal carcinoma [26]. Salinomycin (an antibiotic) is known to suppress cancer stemness manifestations and inhibit or reverse the EMT in cancer cells of different origins [304,305], although there are no data supporting the effects of this drug on the radiosensitivity of hypoxic tumors. It has been established that resveratrol (a natural stilbenoid with antitumor properties) is able to reverse the EMT in nasopharyngeal carcinoma cells and also suppress the pivotal traits of CSCs such as the self-renewal, metabolic reprogramming, resistance to drugs and X-rays and capacity for metastasis formation [306]. Sunitinib, a multityrosine kinase inhibitor and suppressor of HIF-1α, was demonstrated to radiosensitize prostate CSCs and reduce intratumoral hypoxia [97]. In vitro treatments with δ-tocotrienol, a vitamin E family member, abolished the adaptation of prostate CSCs to hypoxia by reducing both the HIF-1α and HIF-2α levels [307]; this might additionally confer radiosensitization, but such a supposition needs confirmation. According to Koh et al. [308], baicalein (a bioflavonoid) was able to reverse the cancer stemness-associated traits in chemo- and radioresistant breast CSC-like cells and, also, sensitize them to Adriamycin, cis-platin and γ-irradiation. In another study [96], the combination of sorafenib and irradiation preferentially eliminated the CSC populations in the breast cancer cell MDA-MB-231 and MCF-7 lines exposed to hypoxia. Besides radiosensitizing breast CSCs, sorafenib suppressed the expression of HIF-1α and MMP2 in MDA-MB-231 cells, thereby impeding their metastatic activity [96]. These findings [96,97,299,302,303,304,305,306,307,308] indicate a theoretical possibility of pharmacologically targeting the EMT and CSCs in hypoxic tumors. In the future, the use of sunitinib or sorafenib may find an application, as these are the clinically approved drugs that may be combined with radiotherapy, while, in model systems, both acted as HIF-1a suppressors and radiosensitizers of hypoxic cancer cells (see Table 1 and [96,97]). However, CSCs and hypoxia-adapted cancer cells may be less susceptible to small-molecule radiosensitizers due to the overexpression and high activity of membrane transporters ABCG2, ABCB1 and others that pump drugs out of the target cell. Even some of the radiosensitizers locally generated from hypoxia-activated prodrugs [169,170] may be pumped out of the CSC by such transporters. Therefore, inhibition of the pumping-drug-out transporters may improve the effects of some small-molecule radiosensitizers toward hypoxic tumors.

In some cases, the radiosensitization of CSCs can be achieved by affecting the expression of certain microRNAs in them. While hypoxia increased the expression levels of HIF-2α and miR-210 in glioma stem cells, the knockdown of miR-210 impaired the manifestations of cancer stemness and induced the differentiation and radiosensitization of the hypoxic CSCs [309]. In a lung cancer model, miR-18a-5p overexpression led to enhanced radiosensitivity in both the total cancer cell population and CD133+ CSC-like cells; the radiosensitizing action was associated with downregulating ATM and HIF-1α expressions [28]. The above EMT-suppressing and CSC-sensitizing effects of resveratrol treatments on nasopharyngeal carcinoma included the induction of miR-145 and miR-200c, being downregulated in untreated CSCs [306]. Exosomal miR-1255b-5p was shown to suppress the EMT in colorectal cancer cells by inhibiting Wnt/β-catenin pathway activation via the inhibition of human telomerase reverse transcriptase (hTERT) [310]. At present, targeting microRNA expression does not look like a simple curative modality; however, “gene therapy” aimed at selective increasing or decreasing the levels of certain microRNAs in hypoxic cancer cells, CSCs and secreted exosomes seems to be a quite realistic and applicable strategy in the future.

Finalizing this subsection, it should be noticed that a unique molecular composition of the surface of CSCs, as well as the peculiarities of their secretory activity, may provide additional options for targeting CSCs in hypoxic niches. CSCs express on their surface a number of specific markers and antigens (e.g., CD44, CD133, integrin β1, HSP90, GRP78, the extracellular domain of ABCG2 and others [142,246,266,267]) that may be used for the antibody-based vector delivery of drugs or radiosensitizers or radionuclides to CSCs for better treating and sensitizing malignancies. Besides, suitable inhibitors of the expression or activities of heparanase and proteinases (MMPs and uPA) secreted by CSCs may beneficially reduce their migration and metastatic potential. Likewise, the inhibitory targeting of the machinery of exosome generation in CSCs [270] may be conducive to the sensitization of hypoxic tumors (see, also, Section 9).

## 8. How Do Epigenetic Regulators Affect the Radioresistance of Hypoxic Cancer Cells?

The term “epigenetic regulation” implies regulatory modulations of a heritable phenotype that are induced without alterations in the genome nucleotide sequence. In mammalian cells, a set of the epigenetic regulators includes microRNAs, long noncoding RNAs and circular RNAs and, also, enzymes (methylases, histone deacetylases, acetyl transferases and others) that modify DNA or protein components of chromatin; all these factors can influence gene expression in a certain way so that they change a cell phenotype. Epigenetic regulation (or, rather, epigenetic deregulation) plays an important role in tumorigenesis and all major processes associated with the pathogenesis of cancer, such as unlimited tumor growth and invasion, the generation of CSC-like cells and metastasis spread, tumor resistance to therapeutics, etc. (reviewed in [311,312]).

Some of epigenetic regulators (first of all, microRNAs) are implicated in the hypoxia-adapting and radioresistance-promoting mechanisms that act in tumor cells undergoing hypoxic stress.

### 8.1. Hypoxia-Responsive MicroRNAs

Prolonged or chronic hypoxia dramatically alters a profile of microRNAs expressed in the involved tumor cells, which contributes to the adaptive changes of their phenotypes [59,313] and the EMT/development of cancer stemness [271,272,314]. The radioresistance of hypoxia-adapted cancer cells also depends on the spectrum of microRNAs whose expression are somehow altered in response to hypoxic stress (reviewed in [315,316,317]). Here, Table 3 summarizes the data and respective references reflecting the involvement of hypoxia-responsive microRNAs in various cellular mechanisms promoting the radioresistance of hypoxic tumors and, also, lists the major molecular targets of those microRNAs. 

The data presented in Table 3 indicate the variety of microRNA-mediated mechanisms and signaling pathways that promote the radioresistance of hypoxic tumors. In hypoxic cancer cells of different origins, certain microRNAs affect the expression/function of HIF-1 [54,58,123], autophagy [54,255,256,257], apoptosis [54,309,319], energy metabolism reprogramming [123] and other hypoxia-responsive events (see Table 3), thereby conferring radioprotection. Hypoxia-increased miR-210 appears to exert radioprotective effects toward both hypoxic non-stem cancer cells and hypoxic CSCs [54,58,309,319]. Notably, microRNAs contained in exosomes secreted by hypoxic cancer cells may also contribute to the radioresistance of hypoxic tumors; therefore, miR-301a from exosomes generated by hypoxic glioma cells conferred radioresistance via the suppression of TCEAL7 expression and activation of the Wnt/β-catenin transcriptional pathway (see [320] and Section 9).

Besides, microRNAs can activate the migration of hypoxia-stressed cancer cells, as was demonstrated for miR-302a in a breast cancer-related model [321]. In that study, the activation of cancer cell migration was suggested to be due to the expression of C-X-C chemokine receptor-type 4 (CXCR-4), which is the target of miR-302a [321]. Moreover, the hypoxia-induced alterations in the microRNA expression of the involved tumor cells promote EMT and the generation of CSC-like cells [271,272,274,314]. The hypoxia-induced decrease in exosomal miR-1255b-5p secretion by colorectal cancer cells leads to an increased hTERT expression, which enhances the telomerase activity and stimulates EMT [310]. Exosomal miR-23a secreted by hypoxic lung cancer cells increases the vascular permeability by targeting the tight junction protein ZO-1, thereby facilitating the transendothelial migration of the cancer cells and metastasis spread [322] (see Figure 2). In hepatocellular carcinoma, hypoxia-induced exosomes were shown to promote cancer cell proliferation and metastasis formation via a miR-1273f transfer, which stimulated Wnt/β-catenin signaling and EMT [323] (see, also, Section 9). This microRNA-dependent emergence of actively migrating and vasculature-penetrating cancer cells may result in the formation of distant metastases and decrease the efficacy of some kinds of radiotherapy toward hypoxic tumors (see Figure 2).

Such a multifaceted involvement of microRNAs in the radioprotective mechanisms of hypoxia-adapted cancer cells implies both the great contribution of these epigenetic factors to the radioresistance of hypoxic tumors and the presence of multiple targets for radiosensitizing. The references cited in Table 3 indicate that, in in vitro models, the microRNA-mediated radioprotection of hypoxic cancer cells can easily be abolished by either the knockdown or overexpression of the respective microRNAs. Similar approaches with knockdown or overexpression may also be used for targeting the radioresistance intrinsic to CSCs (see Section 7.2). The problem is how to make the knockdown or overexpression of microRNAs doable for radiosensitizing patients’ hypoxic tumors. The targeted delivery of certain plasmid- or virus-based gene vectors would enable to manipulate the expression spectrum of microRNAs in hypoxic cancer cells with a view on improving the outcome of radiotherapy. Probably, the methods of electroporation or special microcarriers (e.g., liposomes or nanoparticles) would be exploited for such microRNA-targeting radiosensitization.

Alternatively, cell-permeable agents may be found or created that will be able to target microRNA expression in hypoxic cancer cells, thus radiosensitizing them. For example, oleuropein, a glycosylated seco-iridoid from green olives, was shown to act as a radiosensitizer toward nasopharyngeal carcinoma both in vitro and in vivo; the radiosensitizing action was associated with the inhibition of HIF-1α/miR519d/p53 and the DNA damage-regulated protein 1 (PDRG1) pathway in oleuropein-treated cancer cells [99]. In another in vitro/in vivo model, oleuropein conferred the radiosensitization of ovarian cancer cells by recovering the HIF-1-suppressed expression of miR-299 in them [60]. Kim et al. [71] combined atorvastatin and radioimmunotherapy with ^131^I-rituximab in Raji xenograft murine models. In that study, atorvastatin was found to induce the miR-346-mediated inhibition of HIF-1α in Raji cells, followed by the suppression of VEGF expression and VEGF-stimulated angiogenesis, which elevated the efficacy of radioimmunotherapy toward the lymphoma xenografts growing in mice [71]. The described effects of oleuropein and atorvastatin allow one to suggest the possibility of selectively manipulating the expression/activities of microRNAs in hypoxic cancer cells by means of small-molecule compounds or some pharmacological agents.

### 8.2. Long Noncoding RNAs and Circular RNAs

Long noncoding RNAs (lncRNAs) participate in the formation of a hypoxic phenotype of cancer cells and seem to be one of the factors defining the tumor response to therapeutics [324]. In particular, HOTAIR, an oncogenic lncRNA overexpressed in human cervical cancer HeLa and C33A cells, rendered them radioresistant via the upregulation of HIF-1α expression [30]. It was found in a non-small cell lung cancer model that lncRNA FAM201A competitively targets miR-370, thereby upregulating HIF-1α and EGFR in cancer cells and decreasing their radiosensitivity [31]. The lncRNA PVT1 expressed in nasopharyngeal carcinoma cells was shown to promote their radioresistance by activating KAT2A acetyltransferase and stabilizing HIF1α [32].

The radioresistance of hypoxic tumor cells may also depend on the antisense transcript of HIF-1a (AHIF) expression. This lncRNA became upregulated in glioma multiforme cells after radiotherapy and the knockdown of AHIF increased their radiosensitivity by promoting apoptosis [325,326]. Notably, exosomes generated by AHIF-overexpressing glioma multiforme cells were conducive to tumor invasion and radioresistance, whereas exosomes generated by the AHIF-knockdown glioma cells suppressed those phenomena [326].

As for hypoxia-responsive lncRNAs, the long, intergenic, noncoding RNA (lincRNA)-p21 expression was demonstrated to become elevated in SMMC7721 hepatoma and U251MG glioma cells in response to hypoxia or X-ray irradiation; the knockdown of lincPNA-p21 caused G2/M phase arrest, stimulated apoptosis and impaired autophagy via the HIF-1/Akt/mTOR/P70S6K pathway in the hypoxic cancer cells of both lines, which became radiosensitized [29].

Circular RNAs (circRNA) are also involved in the adaptive and radioprotective mechanisms acting in cancer cells under hypoxia. So, circRNF20 promoted the proliferation and Warburg effect in breast cancer cells by ensuring the HIF-1α-mediated expression of HK2 in them [125]. In human hepatoma cells, the knockdown of hypoxia-responsive circRNA ZNF292 resulted in an increased SRY (sex-determining region Y)-box 9 (SOX9) nuclear translocation, followed by inhibition of the Wnt/β-catenin pathway; eventually, this led to suppression of both the proliferation and enhancement of radioresistance in hypoxic hepatoma cells in vitro and in vivo [327]. Zhao et al. [126] established a role of circABCB10 in the radiation response of breast cancer cells: circABCB10 regulates miR-223-3p and a target of the latter, profillin-2 (an actin-binding protein), thus facilitating glycolysis and enhancing the radioresistance. CircABCB10 knockdown increased the radiosensitivity of breast cancer cells by suppressing glycolysis via the miR-223-3p/PEN axis; besides, circABCB10-conferred radioresistance can be overcome by the known glycolysis inhibitor 2-deoxyglucose [126].

The hypoxia-responsive involvement of lncRNAs [273,328,329] and circRNAs [330,331] in the mechanisms of EMT and cancer stemness can also be interpreted as a contribution of these epigenetic regulators to the radioresistance of hypoxic tumors, since CSC-like cells generated in hypoxic niches are radioresistant and also have an ability to form distant metastases, thus becoming unavailable for radiotherapeutic exposure by being focused on the primary tumor (see Section 7.1 and Figure 2).

Clinically applicable tools for manipulating lncRNAs and circRNAs in cancer cells remain to be developed. The above encouraging effects achieved with knockdown allow one to suggest the use of “gene therapy”—namely, plasmid- or virus-based vectors expressing siRNA or shRNA—or antisense constructs for selectively targeting certain lncRNAs and circRNAs in tumors. Obviously, this modality will require the usage of specially prepared microcontainers (e.g., liposomes or nanoparticles) and/or approved techniques of electroporation for the delivery of suitable gene vectors to target malignancies. In the future, natural or synthetic small-molecule compounds may be found that will modulate the expression/functions of lncRNAs or circRNAs in hypoxic tumors to sensitize them to therapeutics.

### 8.3. Enzymes Participating in Epigenetic Regulation of Cancer Cell Responses to Hypoxia and Radiation

Intratumoral hypoxia can cause specific alterations in the regulatory modifications (methylation, hypermethylation, acetylation and others) of chromatin in the involved cancer cells that may affect their radioresistance and adaptive capacity [15,16,17,112,316]. Accordingly, certain intracellular enzymes, such as methyltransferases and demethylases, acetyltransferases and deacetylases, and others that catalyze the (de)modification of chromatin in hypoxia-stressed cancer cells, can be considered as potential targets for radiosensitizing hypoxic tumors.

In particular, histone demethylase JMJD2B was shown to promote the resistance of hypoxic gastric cancer cells to γ-irradiation; the radioprotective function of JMJD2B was suggested to be due to the JMJD2B-mediated upregulation of cyclin A1 (CCNA1), while JMJD2B was suggested to be a therapeutic target to overcome the hypoxia-induced radioresistance of tumors [332]. It was demonstrated in another study that emodin (a plant-derived polyphenol with antitumor properties) attenuates the hypoxia-induced radioresistance in hepatocellular carcinoma HepG2 cells via the inhibition of JMJD2B in them [333]. Thus, histone demethylase JMJD2B does seem to be a druggable target for radiosensitizing hypoxic tumors.

As a separate family of epigenetic regulators, histone deacetylases (HDAC) comprise several classes of enzymes that remove acetyl groups from ε-N-acetyl-lysines in histones and nonhistone proteins. HDAC are known to regulate chromatin remodeling, the cell cycle, gene transcription, metabolism, signaling pathways, DNA repair, autophagy and other vital processes in the cell. The functioning of HDAC in cancer cells is necessary for their viability and malignant growth, so that the inhibitors of some members of the HDAC family may be used as anticancer drugs or sensitizers for better targeting tumors [334,335]. As follows from several publications, some inhibitors of HDAC are capable of radiosensitizing hypoxic cancer cells. In particular, treatments with the HDAC inhibitor vorinostat radiosensitized hypoxic colorectal carcinoma cells both in vitro and in xenograft models [336]. Likewise, vorinostat radiosensitized the intrinsically radioresistant prostate cancer DU145 cells under both hypoxic and normoxic conditions [337]. In an in vitro model with human cervical carcinoma, the HDAC inhibitor trichostatin A was shown to be a potent radiosensitizer for HeLa cells under hypoxic conditions; the radiosensitizing effect of trichostatin A was correlated with the downregulation of the expression of HIF-1a and VEGF in hypoxic cancer cells [68].

It should be mentioned that some mechanisms of the HDAC-mediated regulation of cancer cell responses to hypoxia and irradiation involve HSPs, and these mechanisms seem druggable [19,184]. For example, pretreatments of hypoxic hepatoma Hep3B cells with the inhibitor of HDAC5 LMK235 abolished the hypoxia-responsive deacetylation of HSP70 by HDAC5, which subsequently suppressed the HSP70-dependent HIF-1α–HSP90 interactions, nuclear translocation of HIF-1α and expression of HIF-1 targets genes such as GLUT1 and carbonic anhydrase 9 [19]. Taking into consideration that the hypoxia-responsive HIF-1-mediated gene transcription promotes an enhanced radioresistance in hypoxic cancer cells (see Section 2.1 and Section 2.2), one can suggest that the inhibitors of HDAC5 are able to sensitize hypoxic tumors to radiation exposure. The inhibitor of HDAC6 LBH589 (Panobinostat) was shown to prevent the deacetylation of HSP90 in tumor cells, which led to the radiosensitization of them, owing to the chaperone inactivation and failure of the HSP90-dependent radioprotective mechanisms [184]; it seems likely that LBH589 (an approved anticancer drug) is applicable for the radiosensitization of hypoxic tumors as well.

Interestingly, several members of the HDAC family are implicated in the development/maintenance of cancer stemness and fulfilment of the EMT program, while some HDAC inhibitors are suggested to be used against CSCs [338,339]. If HDAC inhibitors are capable of suppressing the EMT-associated generation of radioresistant and actively migrating CSC-like cells in hypoxic niches, such agents may sensitize hypoxic tumors to radiotherapy.

Sirtuin 1 (Sirt1) is a nicotinamide adenine dinucleotide (NAD)-dependent deacetylase that is a member of the HDAC family and participates in cancer-related epigenetic regulation by catalyzing the deacetylation of histones and non-histone proteins [340]. Joo et al. [23] found that hypoxia-activated Sirt1 directly binds to HIF-1α and performs its deacetylation in hypoxic cancer cells, which leads to both HIF-1α stabilization/accumulation and the expression of HIF-1α target genes such as VEGF, GLUT1 and MMP2; therefore, tumor cell adaptation to hypoxia and tumor invasion requires the hypoxia-responsive activation of Sirt1. In models with hepatoma HepG2, the elevated radioresistance of hypoxic cancer cells is associated with the hypoxia-induced deacetylation/stabilization of c-Myc by Sirt1 [341,342]. Notably, nicotinamide (an inhibitor of Sirt1) enabled the radiosensitization of hypoxic HepG2 cells by disturbing the Sirt1-mediated regulation of the accumulation/degradation of c-Myc under hypoxia [342]. Although nicotinamide (vitamin B3) is bioavailable and lowly toxic, respective trials are needed to assess the in vivo feasibility of such an approach toward hypoxic tumors.

It is expected that further progress in the development of clinically applicable tools for modulating the epigenetic regulation of tumors will help radiotherapists to overcome the radioresistance of hypoxia-adapted cancer cells.

## 9. Hypoxia-Induced Exosome Generation by Tumor Cells

Within hypoxic niches, the exosome-mediated interrelations between cancerous and stromal cells are believed to promote both cancer cell adaptation to hypoxia and rearranging the cancer cell microenvironment toward the facilitation of tumor growth [343,344]. Exosomes produced by hypoxic cancer cells appear to play critical roles in the tumor angiogenesis, invasion, generation of CSC-like cells, metastasis spread and tumor resistance of the immune system and therapeutics (see [270,343,344] and Figure 1 and Figure 2). The present section reviews a contribution of hypoxia-induced exosomes to the radioresistance of hypoxic tumors and, also, some approaches to attenuating this contribution or exploiting special exosomes against the radioresistant hypoxic tumors.

### 9.1. Hypoxia-Induced Exosomes Can Promote the Radioresistance of Hypoxic Tumors

It is thought to be an established fact that exosomes generated by hypoxic tumor cells act as intercell transmitters of tolerance to hypoxia; this is due to the specific content of those exosomes—namely, certain proteins and RNAs that trigger the HIF-1α-mediated transcription response, UPR, autophagy and angiogenic signaling in recipient cells (reviewed in [344]).

Some of the exosome-induced effects on recipient cells may promote the radioresistance of hypoxic tumors. Exosomal proteome profiling was proposed to identify the protein component(s) responsible for hypoxia-induced radioresistance in breast cancer [345]. However, the impact of exosomal microRNAs seems no less important. In particular, the miR-301a-containing exosomes produced by hypoxic glioblastoma cells were shown to be involved in the acquisition of radioresistance; mechanistically, exosomal miR-301a inhibited the TCEAL7 expression and, hence, activated the Wnt/β-catenin transcriptional pathway, which was otherwise downregulated by TCEAL7 [320]. The revealed exo-miR-301a/TCEAL7-signaling axis has been suggested as a potential target for the sensitization of glioblastoma multiforme to radiotherapy [320]. It was elucidated in another study that the overexpression of AHIF (antisense transcript of HIF-1α) found in glioblastoma multiforme cells/tissue is the endogenous factor that defines the generation of radioresistance-spreading exosomes [326]; consequently, inhibiting AHIF overexpression in glioblastoma multiforme may increase the radiosensitivity of this type of tumor.

Importantly, intratumoral hypoxia induces exosomes, promoting cancer cell migration, EMT and the propagation of a CSC-like (prometastatic) phenotype [270,343,344]. As the EMT and accumulation of actively migrating CSC-like cells decrease the efficacy of radiotherapy toward hypoxic tumors (see Section 7 and Figure 1, Figure 2, Figure 4 and Figure 5), the important question is: which hypoxia-associated peculiarities in the exosome generation are conducive to the development/maintenance of cancer stemness? Ramteke et al. [346] compared exosomes generated by hypoxic and normoxic prostate cancer cells (LNCaP and PC3 lines) and found that the hypoxic cancer cells secrete smaller exosomes, which manifested a higher MMP activity and contained more (160 vs. 62) various proteins and greater amounts of tetraspanins (CD63 and CD81); HSP90 and HSP70; annexin II and signaling proteins (TGF-β2, TNF1α, IL6, TSG101, Akt, ILK1 and β-catenin) as compared to exosomes secreted by normoxic cancer cells. Notably, such exosomes produced by hypoxic prostate cancer cells increased the stemness and invasiveness of naïve LNCaP and PC3 cells under coculturing; the observed effects were explained by the exosome-induced remodeling of the epithelial adherens junction pathway in recipient carcinoma cells [346]. In a model with colorectal cancer, hypoxia caused a decrease in exosomal miR-1255b-5p secretion by cancer cells; being liberated from the inhibitory influence of miR-1255b-5p, the Wnt/β-catenin pathway became activated and enhanced the hTERT expression, which led to an increase in telomerase activity and realization of the EMT program [310]. In another model, exosomes secreted by hypoxic hepatocellular carcinoma cells promoted EMT and metastases by transferring miR-1273f, which activated the Wnt/β-catenin pathway in the recipient cancer cells [323]. As for exosomal long noncoding RNAs, the hypoxia-induced transmission of exosomal lncRNA PCGEM1 was found to promote cell migration, invasiveness and metastatic potential in gastric cancer cells by maintaining the stability of SNAI1, which induced EMT [329].

There are publications demonstrating that hypoxia-induced exosomes can enhance tumor cell migration, invasion and metastasis formation, while EMT induction and cancer stemness were not studied there [322,347,348]. MiR-23a-containing exosomes derived from hypoxic lung cancer cells increased the vascular permeability by targeting the tight junction protein ZO-1 in the vascular endothelium, which could facilitate cancer cell penetration into the blood vessels and the dissemination of metastases [322]. Huang et al. [347] reported that exosomes generated by hypoxic colorectal cancer cells transfer and introduce Wnt4 to normoxic cancer cells to activate β-catenin signaling and confer with them a prometastatic phenotype. Both hypoxia and γ-irradiation were shown to induce the secretion of exosomes by lung carcinoma A549 cells, and the exosomal protein angiopoietin-like 4 was identified as a factor activating cancer cell migration and invasion [348]. Although, in those models [322,347,348], the hypoxia-driven induction of EMT and cancer stemness were not explored, the revealed facts of hypoxia- or irradiation-activated migration and invasion of cancer cells allow one to suggest the scenario shown in Figure 2; if such a scenario is realized in a patient’s body, this may diminish the efficacy of some regimens of radiotherapy.

### 9.2. Exosomes as Targets or Tools to Attenuate the Radioresistance of Hypoxic Tumors

The above reference [310] indicates that exosomal miR-1255b-5p can inhibit the Wnt/β-catenin pathway in recipient cancer cells, thereby suppressing the hTERT expression, telomerase activity and EMT, which are depended on Wnt/β-catenin signaling; however, those miR-1255b-5p-mediated beneficial effects are abolished by hypoxia, which reduces exosomal miR-1255b-5p secretion. Sometimes, it is possible to artificially prevent the formation and secretion of cancer-promoting exosomes; for example, as a result of AHIF knockdown in glioblastoma multiforme cells, their exosomes initially spreading the radioresistance and invasiveness became converted into exosomes that inhibited those traits [326]. Hypothetically, the development of methods for suppressing exosome secretion by hypoxic cancer cells or the beneficial recomposing of the cargoes of such exosomes before their secretion or the early neutralizing of cancer cell-derived extracellular exosomes may provide oncologists with novel, unique tools for effectively targeting hypoxic tumors.

Intriguingly, there are approaches to the use of hypoxia-induced exosomes as tools for better targeting tumors. Therefore, in a model of triple-negative breast cancer, exosomes were released by human breast cancer MDA-MB-231 cells being under hypoxic or normoxic conditions and exposed to X-ray irradiation or not; the isolated exosomes were modified to carry superparamagnetic iron oxide (a label for monitoring in vivo) and Olaparib (a PARP inhibitor and radiosensitizer) [349]. It was found that the hypoxic cancer cells preferentially incorporated exosomes from the hypoxic cancer cells as compared with the other exosome variants; the therapeutic potential of the Olaparib-loaded exosomes was manifested through enhancing apoptosis and slowing down the tumor xenograft growth in nude mice. Such an approach could be used to monitor the modified exosomes in vivo and deliver radiosensitizers to hypoxic tumors [349]. Overall, the idea to use exosomes for the curative targeting of tumors seems very exciting; in recent years, various strategies have been actively developed to exploit artificially engineered exosomes as biogenic nanoparticles for therapy (for a review, see [350]). While drug transport to the hypoxic regions of tumors has met serious impediments [106], the use for this purpose of specially prepared (drug-loaded) exosomes may be rather effective.

## 10. Conclusions and Perspectives

Taking into consideration all the reviewed data, one can conclude that the tumor cell response to chronic or prolonged hypoxia is a complex, multifactor and multilevel process that includes dramatic alterations in gene expression, signaling pathways, energy metabolism, epigenetic regulation, the work of chaperones, autophagy, secretory activity and other stress-sensitive mechanisms of the involved cancer cells. Many of those alterations are cancer cell-adapting mechanisms that contribute to the enhanced radioresistance of hypoxic tumors (see Figure 1, Figure 2, Figure 4 and Figure 5). Some of these radioprotective mechanisms appear to act under both chronic and cycling hypoxia in tumors.

From the data reviewed, it follows that the radiosensitization of hypoxia-adapted cancer cells is not a simple problem at all, because, in vivo, too many various factors may somehow affect the outcome of radiotherapy, even including a factor such as the circadian rhythm [57]. The authors of the review believe it is important that the stress-responsive formation of a hypoxia-resistant cancer cell phenotype (and a CSC-like phenotype) occurring in poorly vascularized tumor regions or in microsphere-embolized tumors is accompanied by the enhancement of tumor radioresistance.

Indeed, the relevant references considered in Section 2, Section 3, Section 4, Section 5, Section 6, Section 7, Section 8 and Section 9 show that all the listed responses of cancer cells to hypoxia can promote their high radioresistance, which is conferred through involving different cellular mechanisms and regulatory pathways, such as HIF-1-mediated gene transcription, metabolic reprogramming, heat stress response and UPR, autophagy, EMT and others (Figure 1, Figure 2, Figure 4 and Figure 5). Importantly, all those mechanisms triggered in hypoxia-adapted cancer cells are interrelated and can attenuate the effectiveness of some strategies suggested for better targeting hypoxic tumors. For instance, the artificial oxygenation of hypoxic tumors before radiation treatments has a rationale, because such an approach enables, first, the solving of the problem of the “oxygen effect” by adding a factor of ROS that is important for low-LET (photon) radiation and, second, to trigger the oxygen-sensitive degradation of HIF-1α, thus downregulating the HIF-1α-dependent radioprotective mechanisms (reviewed in Section 2.1). However, the hypoxia-induced reprogramming of cancer cells results in intracellular GSH accumulation and a better readiness of the DNA break repair system; these hypoxia-responsive mechanisms enhance the antioxidative and radioprotective potential of hypoxic tumors. Besides, CSC-like cells generated in hypoxic niches are also known to be resistant to ROS, apoptosis and radiation exposure (see Section 7). All this appears to impair the radiosensitizing effect of tumor oxygenation performed immediately prior to irradiation, while some pretreatments aimed at the GSH depletion and the reversion of hypoxia-resistant and stem-like cancer cell phenotypes may enhance the oxygenation-conferred radiosensitization. Another example: radiotherapy with proton beams or carbon ion beams is thought to be an effective strategy to treat hypoxic tumors, as it may induce a type of DNA lesions less addicted to the presence of oxygen [4] and rather successfully kills CSCs [65,287,288,289,295,296,297,298,299]. Nevertheless, hypoxia-adapted cancer cells may have some defensive potential against such treatments; in particular, HIF-1α- and HSP90-mediated radioprotective mechanisms appear to be able to impair the efficacy of carbon ion beam therapy, while the inhibition of HIF-1α and/or HSP90 activity may confer the radiosensitization [65,182,183]. Likewise, HSP-inhibiting pretreatments may also improve the radiosensitizing action of local hyperthermia toward hypoxic tumors (see Section 4.5). The development and use of hypoxia-activated prodrugs generating cytotoxic agents or radiosensitizers under hypoxic conditions seems to be a very promising approach to targeting hypoxic tumors [169,170]. Meanwhile, hypoxia-adapted cancer cells and CSC-like cells arising in hypoxic niches often acquire multidrug resistance due to the overexpression of membrane transporters ABCB1 and ABCG2; if so, the effectiveness of some prodrugs (and their derivatives) may be reduced through the pumping-drug-out activity of the membrane transporters. Cotreatments aimed at reversing the phenomenon of drug resistance in hypoxia-adapted cancer cells and CSCs would enhance the sensitizing effects of some hypoxia-activated prodrugs. Additionally, special strategies should be aimed at facilitating drug transporting to the hypoxic regions of target tumors [106].

The large dataset considered in this review demonstrated numerous more-or-less successful attempts to overcome the radioresistance of hypoxic tumors by means of various agents and exposures. It is obvious that the problem cannot be solved using a single agent, and an integrated approach is needed herein. Based on all those data and reasonings, one can suggest that in silico modeling would help to select the optimal combinations of drugs and/or pretreatments that would effectively sensitize hypoxic tumors to radiotherapy, including immunoradiotherapy and radioembolization. The authors of this review suppose that further progress in the radiosensitization of hypoxic tumors will be achieved under combining high-LET radiation exposure and the use of special microcarriers for the targeted delivery of a set of the selected radiosensitizers and anti-CSC agents to hypoxic regions of malignancies. It is expected that the use of nanoparticles or artificially prepared exosomes as vehicles to introduce into hypoxia-adapted cancer cells (and CSCs) as suitable radiosensitizers, along with certain tools of “gene therapy”, e.g., siRNAs or antisense-based constructs, or modulators of epigenetic regulation will enable to overcome the radioresistance of hypoxic tumors.

Combining radiotherapy with immunotherapy can also be delineated as a promising approach to better targeting hypoxic tumors. While intratumoral hypoxia compromises the anticancer immune responses, it was suggested that immune checkpoint inhibitors are able to activate cytotoxic T cells within hypoxic regions of irradiated tumors, thus enhancing the post-radiation immunogenic death of cancer cells [351]. Besides, immunotherapy-induced T-cell activation within hypoxic tumors may increase the efficacy of radiotherapy through the mechanisms that normalize the tumor vasculature and inflow of oxygen-enriched blood [352].

## Figures and Tables

**Figure 1 cancers-13-01102-f001:**
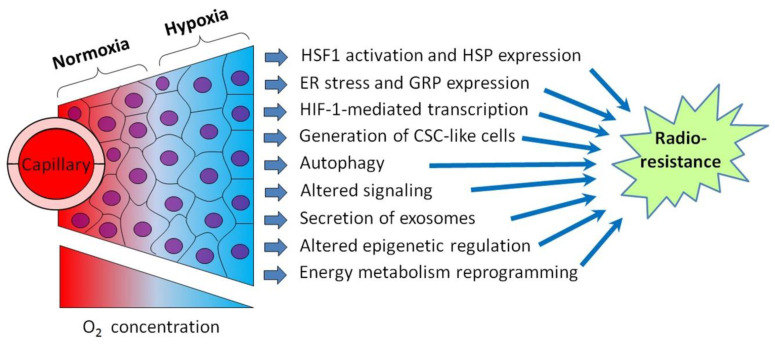
Schematic illustration showing intratumoral hypoxia and major hypoxia-induced cancer cell responses promoting the radioresistance of hypoxic tumors. Abbreviations: HSF1—heat shock factor 1, HSP—heat shock protein, ER—endoplasmic reticulum, GRP—glucose-regulated protein, HIF-1—hypoxia-inducible factor-1 and CSC—cancer stem cell. Tumor hypoxia is usually the result of several factors besides the radial gradient of the oxygen (O_2_) concentration (see [6,7] for a review).

**Figure 2 cancers-13-01102-f002:**
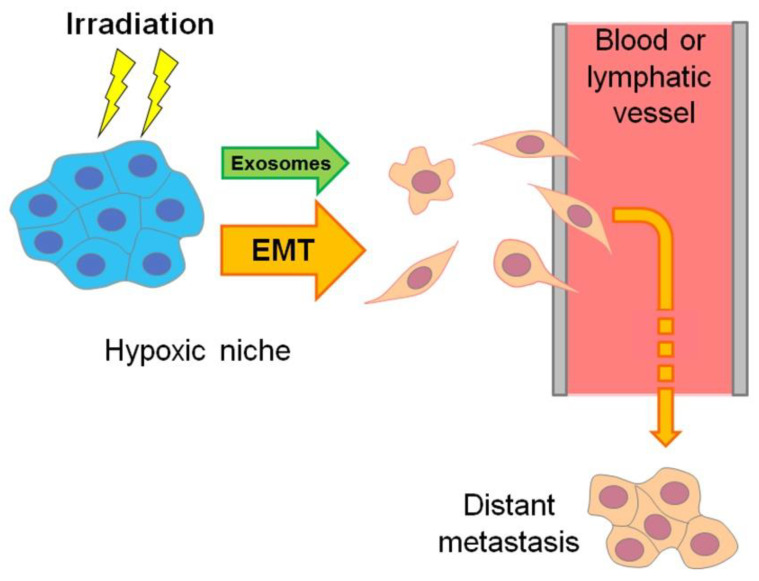
Scheme delineating the hypoxia-associated emergence of actively migrating invasive cancer Cells, which may flee radiotherapeutic exposure by penetrating the vasculature and forming distant metastasis. This hypoxia-driven mechanism is mainly realized through exosome generation and epithelial–mesenchymal transition (EMT) or similar phenotypic modulations toward cancer stemness (see Section 7 and Section 9).

**Figure 3 cancers-13-01102-f003:**
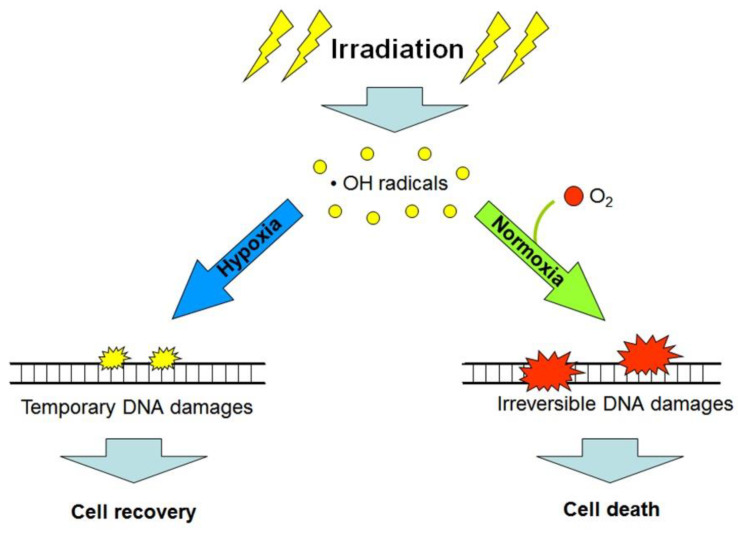
Simplified scheme explaining the “oxygen effect”: low linear energy transfer (LET) radiation exposure in the presence of oxygen (O_2_) produces greater amounts of non-reparable and lethal DNA damages in target cells, as compared with analogous exposure under hypoxic or anoxic conditions (see the text and references [4,10]).

**Figure 4 cancers-13-01102-f004:**
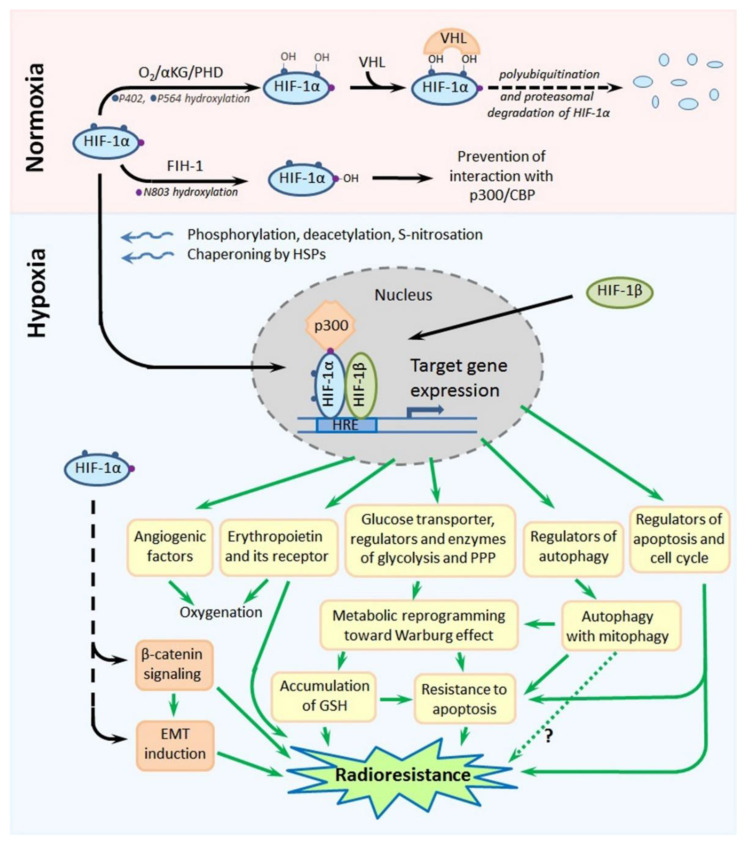
Scheme showing the major mechanisms of regulation of the stability and functional activity of HIF-1α and, also, the major HIF-1α-mediated pathways leading to the radioresistance of hypoxia-adapted cancer cells. Details are given in the text. Abbreviations: HIF-1α—hypoxia-inducible factor-1α, αKG—α-ketoglutarate, PHD—prolyl-4-hydroxylase, VHL—von Hippel-Lindau, FIH-1—factor inhibiting HIF-1, CBP—CREB-binding protein, HIF-1β—hypoxia-inducible factor-1β, HRE—hypoxia-responsive element, GSH—reduced form of glutathione and EMT—epithelial–mesenchymal transition.

**Figure 5 cancers-13-01102-f005:**
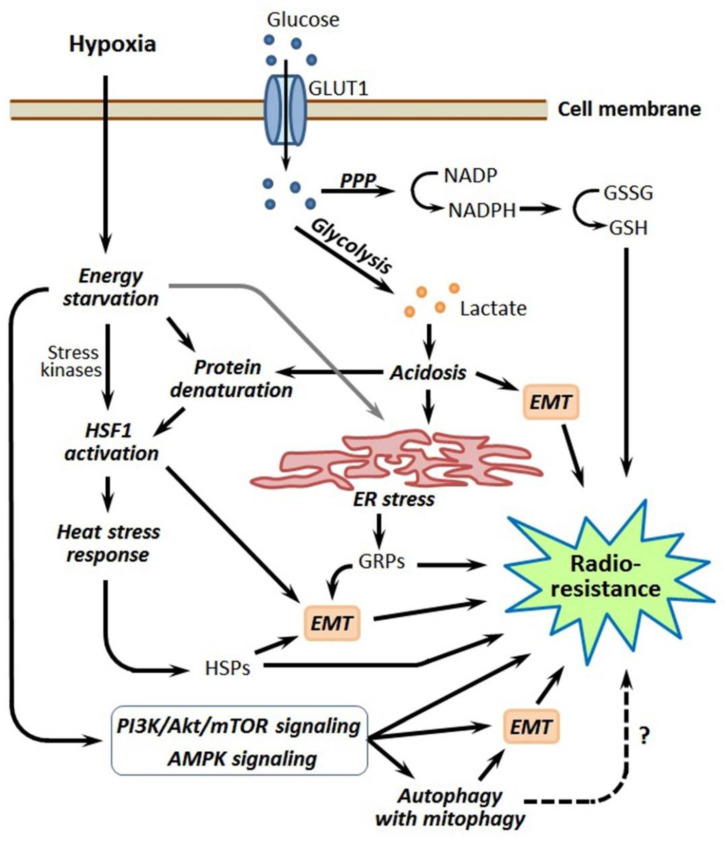
Simplified scheme that links hypoxia-induced energy metabolism reprogramming with other pathways, leading to the radioresistance of hypoxia-adapted cancer cells. Details are given in the text. Abbreviations: GLUT1—glucose transporter 1, PPP—pentose phosphate pathway, NADP and NADPH—nicotinamide adenine dinucleotide phosphate and its reduced form, GSSG and GSH—oxidized and reduced forms of glutathione, EMT—epithelial–mesenchymal transition, HSF1—heat shock factor 1, ER—endoplasmic reticulum, GRPs—glucose-regulated proteins, HSPs—heat shock proteins, PI3K—phospho-inositide 3-kinase, mTOR—mammalian target of rapamycin and AMPK—AMP-activated protein kinase.

**Table 1 cancers-13-01102-t001:** Small molecule agents impairing the HIF-1-based radioresistance of tumor cells.

Agent	Tumor	Molecular Targets	Achieved Effects
Nelfinavir	Head and neck squamous cell carcinoma, lung cancer [66]	HIF-1α and VEGF ^1^ expression	Decreased HIF-1α and VEGF levels, suppressed angiogenesis, radiosensitization
PX-478	Prostate cancer [67]	HIF-1α expression	Decreased HIF-1α level, protracted DNA repair, radiosensitization
Trichostatin A	Cervical carcinoma [68]	HDAC ^2^ activity, HIF-1α and VEGF expression	Decreased HIF-1α and VEGF levels, radiosensitization
Saikosaponin-D (or with PX-478)	Hepatoma [69]	HIF-1α expression	Decreased HIF-1α level, upregulated p53 and Bax, downregulated Bcl-2, increased radiosensitivity
NVP-BEZ235	Breast cancer [37], endometrial cancer [38]	PI3K/mTOR signaling pathway	Suppressed HIF-1α, protracted DNA repair, enhanced apoptosis
Atorvastatin	Prostate cancer [70], Burkitt’s lymphoma [71]	HIF-1α expression miR-346/HIF-1α/VEGF	Downregulated HIF-1α, increased radiosensitivity, anti-angiogenesis
Berberine	Nasopharyngeal [72] and prostate [73] carcinomas	HIF-1α and VEGF	Inhibition of upregulation of HIF-1α and VEGF, radiosensitization
Melittin	Squamous cell carcinoma [74]	HIF-1α and VEGF expression	Reduced expression of HIF-1α and VEGF, increased radiosensitivity
BAY-84-7296	Squamous cell carcinoma [75]	HIF-1α activity	Decreased nuclear translocation of HIF-1α, reduced radioresistance
YC-1	Squamous cell carcinoma [76], cervical carcinoma [77], breast cancer and others [78]	HIF-1α activity	Downregulation of HIF-1α, sensitization to γ-radiation [76,77] and neutron capture therapy [78]
Gambogic acid	Nasopharyngeal carcinoma [79]	HIF-1α expression	Decreased HIF-1α expression, cell cycle arrest, enhanced apoptosis
Pachymic acid	Gastric cancer [80]	HIF-1α expression, Bax expression	Decreased HIF-1α expression, increased Bax, radiosensitization
NSC74859	Esophageal cancer [81]	STAT3, HIF-1α and VEGF expression	Reduced levels of HIF-1α and VEGF, increased radiosensitivity
Ursolic acid	Non-small cell lung cancer [82]	HIF-1α expression, endogenous GSH	Suppressed HIF-1α, decreased intracellular GSH levels
Soy isoflavones	Prostate cancer [83], lung cancer [84]	Src/STAT3/HIF-1α pathway, DNA repair	Inhibition of HIF-1-mediated transcription, radiosensitization
Chetomin	Glioma [85], fibrosarcoma [86], osteosarcoma [87]	HIF-1α–p300 interactions	Suppressed HIF-mediated gene transcription, radiosensitization
2-ME2 ^3^	Nasopharyngeal [26] and esophageal [88] carcinomas, melanoma [89]	HIF-1α and VEGF expression, NF-κB, glycolysis	Downregulation of HIF-1α and VEGF, reversed EMT, increased radiosensitivity
Resveratrol analog	Murine breast cancer [90]	HIF-1α and VEGF expression	Decreased HIF-1α and VEGF levels, inhibited angiogenesis and enhanced apoptosis
Acriflavine	Murine breast cancer [91]	HIF-1α expression	Decreased HIF-1α expression, sensitization to radiotherapy
Temsirolimus	Lung cancer [39]	mTOR, HIF-1α	Decreased HIF-1α expression, radiosensitization
17AAG ^4^, deguelin	Lung cancer [21]	HSP90, HIF-1α	Decreased HIF-1α expression, radiosensitization
KNK437	Breast cancer and glioblastoma [92]	Akt signaling, HIF-1α expression	Decreased HIF-1α expression, radiosensitization
Paclitaxel	Hepatoma and lung adenocarcinoma [93]	HIF-1α-dependent bFGF ^5^/PI3K/Akt pathway	Overcoming HIF-1α-induced radioresistance
Bortezomib	Cervical [94] and esophageal [95] carcinomas	HIF-1α and VEGF expression	Downregulated HIF-1α and VEGF, apoptosis and delayed DNA repair, radiosensitization
Sorafenib	Breast cancer [96]	HIF-1α expression	Suppressing HIF-1α expression, elimination of irradiated CSCs
Sunitinib	Prostate cancer [97]	HIF-1α expression	Decreased HIF-1α expression, radiosensitization of CSCs
FM19G11	Glioblastoma [35]	HIF-1α activity	Radiosensitization
Irisquinone	C6 rat glioma [98]	HIF-1α expression	Downregulation of HIF-1α, radiosensitization
Oleuropein	Nasopharyngeal carcinoma [99]	HIF-1α expression, HIF-1α/miR-519d/PDRG1 ^6^ pathway	Decreased levels of HIF-1α and PDRG1, radiosensitization

^1^ Vascular endothelial growth factor (VEGF), ^2^ Histone deacetylase (HDAC), ^3^ 2-Methoxyestradiol (2-ME2), ^4^ 17-N-allylamino-17-demethoxygeldanamycin (17AAG), ^5^ Basic fibroblast growth factor (bFGF), ^6^ p53 and DNA damage-regulated protein 1 (PDRG1).

**Table 2 cancers-13-01102-t002:** Small molecule radiosensitizers targeting the energy metabolism in tumor cells.

Agent	Tumor	Molecular Targets	Achieved Effects
2-deoxyglucose	Glioblastoma [143,144], breast cancer [126]	HK2 ^1^, glycolysis CircABCB10 ^2^, profilin-2	Inhibition of glycolysis and radiosensitization
Apigenin	Laryngeal carcinoma [145]	PI3K/Akt signaling, GLUT1 ^3^ expression	Downregulated GLUT1, increased radiosensitivity
WZB117	Breast cancer [146]	GLUT1 expression, glucose metabolism	Downregulated GLUT1, radiosensitization
2-ME2	Melanoma [89]	HIF-1α, PDK1 ^4^ and GLUT1 expression	Downregulated PDK1 and GLUT1, radiosensitization
BX795	Hepatocellular carcinoma [120]	PDK1 activity	Increased Bax/Bcl-2 ratio and apoptosis, radiosensitization
Dichloroacetate	Melanoma [89] glioblastoma [119]	PDK1 activity	Increased DNA damage and apoptosis, radiosensitization
BAY-84-7296	Squamous cell carcinoma [75]	Mitochondrial complex I, HIF-1α	Downregulated HIF-1α, impaired radioresistance
Butylmalonate	Lung and prostate cancer, glioblastoma [147]	SLC25A10 ^5^	Overcoming radioresistance induced by hypoxia
Metformin and phenformin	Colorectal cancer [148]	Mitochondrial complex I, AMPK	Overcoming radioresistance induced by hypoxia
Anti-parasitic drugs (atovaquone, others)	High-grade glioma [149]	Mitochondrial metabolism	Enhanced radiosensitivity (suggested)

^1^ Hexokinase 2 (HK2), ^2^ Circular RNA ABCB10 (CircABCB10), ^3^ Glucose transporter 1 (GLUT1), ^4^ Pyruvate dehydrogenase kinase 1 (PDK1), ^5^ Mitochondrial dicarboxylate carrier.

**Table 3 cancers-13-01102-t003:** Hypoxia-responsive microRNAs affecting the radioresistance of hypoxic tumor cells.

Hypoxia-Altered MicroRNA Expression	Type of Tumor	Cellular Targets and Induced Effects Leading to the Tumor Radioresistance
Increased miR-155	Lung cancer [318]	Decreased FOXO3A ^1^ expression
Increased miR-210	Hepatoma [319]	Decreased AIFM3 ^2^, protection from apoptosis
	Lung cancer [58]	HIF-1-dependent DNA double-strand break repair
	Glioma stem cells [309]	Decreased MNT ^3^, protection from apoptosis
	Colon cancer [54]	HIF-1α/miR-210/Bcl-2 pathway, protection from apoptosis, enhanced autophagy
Increased miR-21	Lung cancer [123]	Upregulation of HIF-1α, activated glycolysis
Decreased miR-124/miR-144	Prostate cancer [255]	Downregulated PIM1 ^4^, induced autophagy
Decreased miR-30a/miR-205	Prostate cancer [256]	Increased TP53INP1 ^5^, induced autophagy
Increased miR-301a/miR-301b	Prostate cancer [257]	Decreased NDRG2 ^6^, elevated autophagy
Decreased miR-519d	Nasopharyngeal cancer [99]	Decreased levels of HIF-1α and PDRG1 ^7^
Exosomal miR-301a	Glioma [320]	Suppressed TCEAL7 ^8^ and activated Wnt/β-catenin pathway

^1^ A protein product of the Forkhead box O3 (FOXO3) gene, a transcriptional factor. ^2^ Apoptosis-inducing factor mitochondria-associated 3 (AIFM3), a protein effector of apoptosis. ^3^ A protein product of the Myc antagonist MAX’s Next Tango (MNT) gene. ^4^ Proto-oncogene serine/threonine-protein kinase Pim-1. ^5^ Tumor protein P53-inducible nuclear protein 1 (TP53INP1). ^6^ A protein product of the NMYC downstream-regulated gene 2 (NDRG2). ^7^ p53 and DNA damage-regulated protein 1 (PDRG1). ^8^ A protein product of the transcription elongation factor A-like 7 (TCEAL7) gene.

## Data Availability

Data sharing is not applicable to this article.

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
