# Peer review of "Hypoxia-Induced Cancer Cell Responses Driving Radioresistance of Hypoxic Tumors: Approaches to Targeting and Radiosensitizing"

_cancers, 2021, doi:10.3390/cancers13051102_

Round 1
Reviewer 1 Report
A lot of information is given in this extensive review.
It would be nice to see a visual figure when talking about the energetic metabolism or PMT transition.
Author Response
Ref: Revised Version A.E. Kabakov and A.O. Yakimova cancers-1066734
Authors’ responses to Reviewer 1:
Dear Reviewer,
Thank you very much for reviewing and positive estimation of our manuscript. We are fully agreed with your suggestion regarding a visual figure linking the hypoxia-induced alterations in energy metabolism and EMT. Your suggestion coincided with Reviewer 3’ advice – that is why we have added 2 new Figures in the revised version. We think that Figure 5 (see the page 12) is close to what you meant.
With best wishes,
On behalf of both authors,
Alexander Kabakov
Reviewer 2 Report
Labakov E.A and Yakimova A.O reported in this review all the hypoxia induced cellular responses that contribute to the radioresistance of hypoxic tumors. The review is well written and organized and the literature is well updated.
Author Response
Ref: Revised Version A.E. Kabakov and A.O. Yakimova cancers-1066734
Authors’ responses to Reviewer 2:
Dear Reviewer,
Thank you very much for reviewing and positive estimation of our manuscript.
With best wishes,
On behalf of both authors,
Alexander Kabakov
Reviewer 3 Report
The authors tackle an exciting topic that could potentially improve management and hopefully improved outcomes of several cancers.
Abstract should include 2-3 summary sentences for most “hypoxic tumors” that are likely to benefit from radiosensitizing approaches.
The review is very long. It should be shrink a lot (Using scheme could help for that aim). The aim of the review is to provide an overview on the topic. Scheme that depict HIFs, hypoxia and metabolism , HSF1, HSPs, GRPs, Autophagy and their connections with CSCs could be helpful.
At the same time, sentences should be concise and thoroughly enlighten the topic. For instance:
- Paragraph 6.1, 6.2 and 6.3 could be combined. Deleting Line 1064 to 1076 will not affect the review quality. Line 1076-1078 doesn’t explained how does chloroquine target autophagy and help to radiosensitize hypoxic tumors.
-Right to the target and more concise and shrink: line 674 to 692
-Too much repetition: line 801 to 811 and line784-790. Already developed in section with HSP90, HSP 27 and HSF1
-Line 758-760: should be delete. Clinical trial (phase I,II and III) with selective inhibitor of HSP90 need to be run to get an approval in clinical practise.
Line 841: replace “specially recognizing” by “targeting”.
Delete Line 849-850: from “There is a probability” … to “hypoxic tumors”.
Line 624-626: repetition
Line 468-480, 624-626, 655-675, 723-799; 823-824: repetition with
Line 1143-1147: repetition. Already introduced in 1103-1124
Line 1152-1157: repetition. Already introduced in 521-532
Line 1176-1186: repetition. Already developed in the section 2.1. 4.2. 4.3 and 5.
Line 1218-1224: has been approached in 1103-1111
Line 1248: replace “facts” by “data”.
Line 1264-1270: repetition. Already developed in the section 4 and 5.
Line 1272: replace “killing or sensitizing” by “targeting”.
Line 1272: replace “created” by “developed”.
Line 1276: replace “was” by “has been”.
Line 1283: replace “publications yet about” by “data supporting”.
Line 1284: replace “was” by “has been”.
Table 1 as well as table 2 should include a column with a type of evidence (preclinical, phase I, Phase II , phase III clinical trail) . It will help to put data into perspective (clinical evidence vs only preclinical evidence)
Line 210: TAT-Lp15 abbreviation definition
Line 411-412: ABCG2, ABCB1 abbreviation definition
Fig 1: abbreviation definitions (HSF1, ER, HSP…) as in the table 3.
Fig 2 and 3: too simple. It would be great to replace them by a scheme that represents in details biological pathways driving radioresistance of hypoxic tumors , for instance: Hypoxia-inducible factors (part 2) and Hypoxia-induced reprogramming of energy metabolism (part 3).
Scheme should represent line (151 to 435) which depict the role of HIF, HSF pathways in EMT and radioresistance. It should also represent role of hypoxia in cell recovery: line (201-202) and line ( 432 to 436 , line 455to 461and line 484 to 489
Fig 4: doesn’t provide any important information.
Line 1299-1301: Data should be put into perspective: Combo of sunitinib or sorafenib results in unacceptably high rates toxicity in some clinical trials ( R.B. Goody et al. /Radiotherapy and Oncology 123 (2017) 234–239237, Phase 1 Trial of Sorafenib and Stereotactic Body Radiation Therapy for Hepatocellular Carcinoma": https://www.sciencedirect.com/science/article/pii/S036030161526796X; Chi KH, Liao CS, Chang CC, Ko HL, Tsang YW, Yang KC, Mehta MP (2010) Angiogenic blockade and radiotherapy in hepatocellular carcinoma. Int J Radiat Oncol Biol Phys 78:188–193; Kao J, Packer S, Vu HL, Schwartz ME, Sung MW, Stock RG, Lo YC et al (2009) Phase 1 study of concurrent sunitinib and image-guided radiotherapy followed by maintenance sunitinib for patients with oligometastases: acute toxicity and preliminary response. Cancer 115:3571–3580), Wuthrick EJ, Curran WJ Jr, Camphausen K, Lin A, Glass J, Evans J, Andrews DW et al (2014) A pilot study of hypofractionated stereotactic radiation therapy and sunitinib in previously irradiated patients with recurrent high-grade glioma. Int J Radiat Oncol Biol Phys 90:369–375).
It would be great to tackle the role of immunotherapy with immune check point inhibitor to radiosensitize hypoxic tumors
Author Response
Ref: Revised Version A.E. Kabakov and A.O. Yakimova cancers-1066734
Authors’ responses to Reviewer 3:
Dear Reviewer,
Thank you very much for reviewing, helpful criticism and instructive editing of our manuscript. We are agreed with many points of your criticism. We can also understand your irritation about repetitions in the text of the manuscript. In our own defense, we can say that some of those repetitions are inevitable and even necessary - they emphasize the interconnection of all the processes considered. In addition, if we consider multifactorial mechanisms that simultaneously affect several cell responses, either factor has to be mentioned in each respective section - otherwise the mechanism will not be explained. Below we respond point-by-point to your remarks. Please note that all new insertions added in the text in response to your suggestions are marked by yellow.
You wrote:
Abstract should include 2-3 summary sentences for most “hypoxic tumors” that are likely to benefit from radiosensitizing approaches.
Authors’ response: Thanks for your suggestion. Preparing the revised version, we have somewhat remade Abstract. We think that the final sentence in new Abstract (marked by yellow) is close to what you recommended.
You wrote:
The review is very long. It should be shrink a lot (Using scheme could help for that aim). The aim of the review is to provide an overview on the topic. Scheme that depict HIFs, hypoxia and metabolism , HSF1, HSPs, GRPs, Autophagy and their connections with CSCs could be helpful.
Authors’ response: Thanks for your criticism and suggestions. We have tried to shrink our manuscript as much as possible (on our opinion). We have removed many “surplus” paragraphs, sentences and words throughout the text. Figure 4 in the old version was removed as well. Instead, we have added 2 new Figures (see Figures 4 and 5 in the revised version) which depict HIF-1 regulation, energy metabolism reprogramming, HSF1, HSPs, GRPs, autophagy and their connections with EMT (CSCs), as you suggested. Please note that, although approximately 2 pages of text were removed from the primary text of our manuscript, the addition of 2 new Figures and decoding of many abbreviations (as you recommended to do) have again increased the volume of our manuscript to 53 pages. Nevertheless, we hope that you will find our revised manuscript as improved because now it contains less text and more schemes.
You wrote:
At the same time, sentences should be concise and thoroughly enlighten the topic. For instance:
- Paragraph 6.1, 6.2 and 6.3 could be combined.
Authors’ response: Thanks, this has been done (see the newly restructured section 6 on pages 22-24 of the revised version).
Deleting Line 1064 to 1076 will not affect the review quality.
Authors’ response: Probably, the numbering of lines does not quite coincide in the manuscript layout opened on your computer and my computer. It seems to us that the indicate fragment of text contains some important information and we prefer to keep it.
Line 1076-1078 doesn’t explained how does chloroquine target autophagy and help to radiosensitize hypoxic tumors.
Authors’ response: Thanks for your remark, now it has been done. Please see the yellow-marked fragment of text on page 24 in the revised version of our manuscript.
-Right to the target and more concise and shrink: line 674 to 692
Authors’ response: Thanks, this fragment of the text has been shortened (see the respective paragraph on page 17 of the revised version).
-Too much repetition: line 801 to 811 and line784-790. Already developed in section with HSP90, HSP 27 and HSF1.
Authors’ response: Thanks, we have tried to maximally shorten these fragments of the text.
-Line 758-760: should be delete. Clinical trial (phase I,II and III) with selective inhibitor of HSP90 need to be run to get an approval in clinical practise.
Authors’ response: Those lines have been deleted (see on page 18 of the revised version).
Line 841: replace “specially recognizing” by “targeting”.
Authors’ response: this has been done, see on the page 19 (marked by yellow).
Delete Line 849-850: from “There is a probability” … to “hypoxic tumors”.
Authors’ response: Those lines have been deleted (see the respective paragraph on the page 19).
Line 624-626: repetition
Line 468-480, 624-626, 655-675, 723-799; 823-824: repetition with
Line 1143-1147: repetition. Already introduced in 1103-1124
Line 1152-1157: repetition. Already introduced in 521-532
Line 1176-1186: repetition. Already developed in the section 2.1. 4.2. 4.3 and 5.
Line 1218-1224: has been approached in 1103-1111
Authors’ response: we have tried to shorten all the text fragments with repetitions as much as possible. But we could exclude all the repetitions because they often reflect the interconnection between different multifactor mechanisms and pathways.
Line 1248: replace “facts” by “data”.
Authors’ response: this has been done, see on the page 27 (marked by yellow).
Line 1264-1270: repetition. Already developed in the section 4 and 5.
Authors response: We have shortened this paragraph (see on the page 27).
Line 1272: replace “killing or sensitizing” by “targeting”.
Line 1272: replace “created” by “developed”.
Line 1276: replace “was” by “has been”.
Line 1283: replace “publications yet about” by “data supporting”.
Line 1284: replace “was” by “has been”.
Authors’ response: Thanks, all these corrections have been done, see on the pages 27, 28 (marked by yellow).
You wrote:
Table 1 as well as table 2 should include a column with a type of evidence (preclinical, phase I, Phase II , phase III clinical trial). It will help to put data into perspective (clinical evidence vs only preclinical evidence).
Authors’ response: Yes, it would be a good idea, if these Tables did contain any data come from clinical trials. But both Tables contain only experimental data obtained in in vitro and in vivo model systems (i.e. cell cultures, murine or rat tumors, xenografts growing in nude mice). We indicate that this is only a set of experimental data (“proof-of-principle”) - see the yellow-marked sentences on the pages 8, 10, 14, and 15. The data on metformin partly supported by the Phase I dose-finding study are discussed below the Table 2 (ref [146]) – see the page 15. Therefore, we suppose that the additional columns in the Tables 1 and 2 is not needed.
You wrote:
Line 210: TAT-Lp15 abbreviation definition
Line 411-412: ABCG2, ABCB1 abbreviation definition
Authors’ response: These abbreviation definitions are now given on the pages 6 and 11 (marked by yellow).
You wrote:
Fig 1: abbreviation definitions (HSF1, ER, HSP…) as in the table 3.
Authors’ response: These abbreviation definitions are now given in a legend to Figure 1 (see on the page 2, marked by yellow).
You wrote:
Fig 2 and 3: too simple. It would be great to replace them by a scheme that represents in details biological pathways driving radioresistance of hypoxic tumors , for instance: Hypoxia-inducible factors (part 2) and Hypoxia-induced reprogramming of energy metabolism (part 3).
Scheme should represent line (151 to 435) which depict the role of HIF, HSF pathways in EMT and radioresistance. It should also represent role of hypoxia in cell recovery: line (201-202) and line ( 432 to 436 , line 455to 461and line 484 to 489.
Authors’ response: Yes, you are correct. We have now added 2 new Figures (Figures 4 and 5 in the revised version). We hope that these new Figures are close to what you meant. (see Figure 4 on the page 5 and Figure 5 on the page 12).
You wrote:
Fig 4: doesn’t provide any important information.
Authors’ response: Yes, you are correct. That Figure has been removed from the revised version.
You wrote:
Line 1299-1301: Data should be put into perspective: Combo of sunitinib or sorafenib results in unacceptably high rates toxicity in some clinical trials ( R.B. Goody et al. /Radiotherapy and Oncology 123 (2017) 234–239237, Phase 1 Trial of Sorafenib and Stereotactic Body Radiation Therapy for Hepatocellular Carcinoma": https://www.sciencedirect.com/science/article/pii/S036030161526796X; Chi KH, Liao CS, Chang CC, Ko HL, Tsang YW, Yang KC, Mehta MP (2010) Angiogenic blockade and radiotherapy in hepatocellular carcinoma. Int J Radiat Oncol Biol Phys 78:188–193; Kao J, Packer S, Vu HL, Schwartz ME, Sung MW, Stock RG, Lo YC et al (2009) Phase 1 study of concurrent sunitinib and image-guided radiotherapy followed by maintenance sunitinib for patients with oligometastases: acute toxicity and preliminary response. Cancer 115:3571–3580), Wuthrick EJ, Curran WJ Jr, Camphausen K, Lin A, Glass J, Evans J, Andrews DW et al (2014) A pilot study of hypofractionated stereotactic radiation therapy and sunitinib in previously irradiated patients with recurrent high-grade glioma. Int J Radiat Oncol Biol Phys 90:369–375).
Authors’ response: Yes, you are correct. A part of suggestions we made are somewhat naïve. Now we have tried to change that fragment of the text and the new version sounds better: see the respective yellow-marked text fragment on the page 28.
You wrote:
It would be great to tackle the role of immunotherapy with immune check point inhibitor to radiosensitize hypoxic tumors.
Authors’ response: Thanks for this good idea! Now we discuss such an opportunity in the final paragraph of the section Conclusion and perspectives of the revised version (see the yellow-marked text fragment on the page 36. A couple of the relevant refs [340] and [341] have been added herein.
So, it seems to us that we have satisfied the most part of your demands on the improvement of our manuscript. We hope that you will find our revised version as the improved one.
With best wishes,
On behalf of both authors,
Alexander Kabakov

Reviewer 4 Report
This manuscript contains many content but need to link between sections.
Authors need to reorganize contents from flow of logics of writing.
Some subtitles need to reflect contents of sections.
for example,
4.1. HSF1
4.2. HSP90
4.3. HSP70
4.4. HSP27
Author Response
Ref: Revised Version A.E. Kabakov and A.O. Yakimova cancers-1066734
Authors’ responses to Reviewer 4:
Dear Reviewer,
Thank you very much for reviewing our manuscript.
You wrote:
This manuscript contains many content but need to link between sections.
Authors’ response: We are afraid that the addition of special text fragments or paragraphs to better link one section to another would sharply increase the volume of our manuscript which is already very large. Please note that the revised version now contains 2 new Figures (Figure 4 and Figure 5) which help to conceptually link the sections with each other (see pages 5 and 12 of the revised version). So, please allow us not to add special (textual) links between the section of our manuscript.
You wrote:
Authors need to reorganize contents from flow of logics of writing.
Authors’ response: We are not agreed, though, that the content in our manuscript is designed without logics. On the contrary, it seems to us that such an order of the sections and subsections is logical and suitable. Please look at our logics herein: we began from HIF-1 (the first section after Introduction) because this factor is the master regulator of the tumor cell response to hypoxia. Accordingly, virtually all other hypoxia-responsive mechanisms (including the radioprotective ones) are somehow related to HIF-1. After the HIF-1-dependent gene transcription and signaling pathways, the energy metabolism seems to be the most significant cellular sensor of hypoxia and it is the next section in our manuscript. The hypoxia-provoked dramatic alterations in the energy metabolism cause heat stress response and HSP induction, endoplasmic reticulum stress and GRP induction, and autophagy in the involved tumor cells (the next 3 sections in our manuscript). All the above hypoxia-induced responses can induce the EMT and generation of CSC-like cells (the section 7) etc. So, please allow us to keep the present content of sections/subsections in our manuscript.
You wrote:
Some subtitles need to reflect contents of sections.
for example,
4.1. HSF1
4.2. HSP90
4.3. HSP70
4.4. HSP27
Authors’ response: Yes, you are correct. We have tried to improve those subtitles according to your advice: see the new variants of subtitles on the pages 16, 17, 18, 20 and 22 (marked by green).
With best wishes,
On behalf of both authors,
Alexander Kabakov

Reviewer 5 Report
In the proposed review the authors described hypoxia and the different mechanism by which low O2 partial pressure may induce chemo and radio-resistances, moreover they analyzed the different strategies to overcome this resistance. Overall the review is comprehensive and interesting both for those approaching the study of radio-resistance in hypoxia and for those already working in this field. This review is one of the most interesting and well-structured ones read recently. In my opinion, the only questionable aspect is the manuscript's length, it would be more suitable as a book chapter, but anyway it is well written and the different parts are harmoniously arranged with each other.
Author Response
Ref: Revised Version A.E. Kabakov and A.O. Yakimova cancers-1066734
Authors’ responses to Reviewer 5:
Dear Reviewer,
Thank you very much for reviewing and positive estimation of our manuscript.
With best wishes,
On behalf of both authors,
Alexander Kabakov
Round 2
Reviewer 3 Report
I am sure that Figure 2 is very helpfull.
Author Response
Authors’ response to Reviewer 3:
You wrote: I am sure that Figure 2 is very helpfull.
Authors’ response: Thank you for your remark (if it is not your sarcasm?). Yes, we believe that Figure 2 is fairly good: it shows an unusual mechanism of the cancer cell radioresistance – when the target cell, thanks to its capacity to migrate, goes away from the zone exposed to irradiation. Besides, this Figure shows the involvement of exosomes what is absent in Figures 4 and 5.
With best wishes,
on behalf of both authors,
Alexander Kabakov
Reviewer 4 Report
Line 138: 2. Hypoxia-inducible factors (HIFs) --> section title is too general. --> please change the title reflecting the contents of section 2.
Author Response
Authors’ response to Reviewer 4:
You wrote: Line 138: 2. Hypoxia-inducible factors (HIFs) --> section title is too general. --> please change the title reflecting the contents of section 2.
Authors’ response: Thank you very much for your helpful remark, you are absolutely correct. In the newly revised version, we give the improved variant of this subtitle: Hypoxia-inducible factors: their regulation and contribution to the tumor radioresistance (see on the page 4, marked by blue). We think that such expanded subtitle better reflects the content of Section 2.
With best wishes,
on behalf of both authors,
Alexander Kabakov